# A single pair of pharyngeal neurons functions as a commander to reject high salt in *Drosophila melanogaster*

Jiun Sang[1†], Subash Dhakal[1†], Bhanu Shrestha[1], Dharmendra Kumar Nath[1], Yunjung Kim[1], Anindya Ganguly[2], Craig Montell[2*], Youngseok Lee[1*]

[1]Department of Bio and Fermentation Convergence Technology, Kookmin University, Seoul, Republic of Korea; [2]Neuroscience Research Institute and Department of Molecular, Cellular and Developmental Biology, University of California, Santa Barbara, Santa Barbara, United States

**\*For correspondence:**
cmontell@ucsb.edu (CM);
ylee@kookmin.ac.kr (YL)

[†]These authors contributed equally to this work

**Abstract** Salt (NaCl), is an essential nutrient for survival, while excessive salt can be detrimental. In the fruit fly, *Drosophila melanogaster*, internal taste organs in the pharynx are critical gatekeepers impacting the decision to accept or reject a food. Currently, our understanding of the mechanism through which pharyngeal gustatory receptor neurons (GRNs) sense high salt are rudimentary. Here, we found that a member of the ionotropic receptor family, *Ir60b*, is expressed exclusively in a pair of GRNs activated by high salt. Using a two-way choice assay (DrosoX) to measure ingestion volume, we demonstrate that IR60b and two co-receptors IR25a and IR76b are required to prevent high salt consumption. Mutants lacking external taste organs but retaining the internal taste organs in the pharynx exhibit much higher salt avoidance than flies with all taste organs but missing the three IRs. Our findings highlight the vital role for IRs in a pharyngeal GRN to control ingestion of high salt.

## eLife assessment

This **valuable** study on the molecular and cellular mechanisms of ingestion avoidance of high salt in insects is focused in scope, but the authors present **convincing** evidence that a specific subset of gustatory receptors in a pair of pharyngeal taste neurons are necessary and sufficient for avoiding ingestion of high salt during feeding. This work will be of interest to *Drosophila* neuroscientists interested in taste coding and feeding behavior.

## Introduction

The sense of taste enables animals to find nutritious food while avoiding potentially harmful substances in their environment. Most animals have evolved sophisticated systems to detect and steer clear of consuming levels of substances that are toxic. Salts such as NaCl are essential for a wide array of physiological functions. However, consumption of excessive salt can contribute to various health issues in mammals, including hypertension, osteoporosis, gastrointestinal cancer, autoimmune diseases, and can lead to death (*Heaney, 2006*; *Jones et al., 1997*; *Luft et al., 1979*; *Sharif et al., 2018*; *Strazzullo et al., 2009*; *Neal et al., 2021*). Therefore, high concentrations of salt are rejected by most animals.

Multiple studies have delved into how Na⁺ is sensed in the *Drosophila* taste system, shedding light on the mechanisms behind the attraction to low salt and aversion to high salt (*McDowell et al., 2022*; *Dweck et al., 2022*; *Dey et al., 2023*; *Lee et al., 2017*; *Jaeger et al., 2018*; *Zhang et al., 2013*; *Nakamura et al., 2002*; *Kim et al., 2017*). The largest taste organs in flies are two bilaterally symmetrical labella, each of which is decorated with 31 gustatory bristles (sensilla). These sensilla are defined

based on size (small, S; intermediate, I; large, L). The I-type sensilla harbor two gustatory receptor neurons (GRNs), while the S- and L-sensilla contain four. These GRNs fall into five classes (A-E) based on their response profiles (*Montell, 2021*). These include Class A GRNs (formerly sugar GRNs; marked by *Gr5a* or *Gr64f*) (*Wang et al., 2004*; *Thorne et al., 2004*), which respond to attractive compounds such as low salt, sugars, glycerol, fatty acids, and carboxylic acids; Class B GRNs (formerly bitter GRNs; marked by *Gr66a*) (*Wang et al., 2004*; *Thorne et al., 2004*), which are activated by high $Na^+$, bitter compounds, acids, polyamines, tryptophan, and L-canavanine; Class C GRNs respond to water (marked by *ppk28*) (*Cameron et al., 2010*); Class D GRNs detect high levels of cations such as $Na^+$, $K^+$, and $Ca^{2+}$ (marked by *ppk23*) (*Jaeger et al., 2018*; *Lee et al., 2018*); and Class E GRNs sense low $Na^+$ levels and pheromones (marked by *Ir94e*) (*Jaeger et al., 2018*; *Taisz et al., 2023*).

Several of the 66 ionotropic receptor (IR) family members function in the sensation of low and high salt. These include IR76b and IR25a, which are IR-co-receptors, and therefore have broad roles in sensing many taste stimuli including low and high $Na^+$ (also referred to in this work as salt) (*McDowell et al., 2022*; *Dweck et al., 2022*; *Lee et al., 2017*; *Jaeger et al., 2018*; *Zhang et al., 2013*), $Ca^{2+}$ (*Lee et al., 2018*), several carboxylic acids (*Stanley et al., 2021*; *Shrestha and Lee, 2021*; *Rimal et al., 2019*), fatty acids (*Tauber et al., 2017*; *Brown et al., 2021*; *Kim et al., 2018*; *Ahn et al., 2017*), amino acids (*Aryal et al., 2022a*), and carbonation (*Sánchez-Alcañiz et al., 2018*). A subset of Class A GRNs as well as glutamatergic Class E GRNs are responsible for sensing low salt (*Dweck et al., 2022*; *Jaeger et al., 2018*; *Zhang et al., 2013*; *Montell, 2021*), and this sensation depends on IR56b working together with the broadly expressed co-receptors IR25a and IR76b (*Dweck et al., 2022*). Conversely, detection of high salt depends on Class B GRNs and Class D GRNs, and IR7c, in conjunction with IR25a and IR76b (*McDowell et al., 2022*). Additionally, two Pickpocket channels, Ppk11, Ppk19, and Sano have been associated with high salt aversion (*Alves et al., 2014*; *Liu et al., 2003*).

In addition to the labellum and taste bristles on other external structures, such as the tarsi, fruit flies are endowed with hairless sensilla on the surface of the labellum (taste pegs), and three internal taste organs lining the pharynx, the labral sense organ (LSO), the ventral cibarial sense organ, and the dorsal cibarial sense organ, which also function in the decision to keep feeding or reject a food (*Chen and Dahanukar, 2017*; *Chen and Dahanukar, 2020*; *Stocker, 1994*; *Nayak and Singh, 1983*; *LeDue et al., 2015*). A pair of GRNs in the LSO express a member of the gustatory receptor family, *Gr2a*, and knockdown of *Gr2a* in these GRNs impairs the avoidance to slightly aversive levels of $Na^+$ (*Kim et al., 2017*). Pharyngeal GRNs also promote the aversion to bitter tastants, $Cu^{2+}$, L-canavanine, and bacterial lipopolysaccharides (*Joseph and Heberlein, 2012*; *Xiao et al., 2022*; *Soldano et al., 2016*; *Choi et al., 2016*). Other pharyngeal GRNs are stimulated by sugars and contribute to sugar consumption (*Chen and Dahanukar, 2017*; *LeDue et al., 2015*; *Chen et al., 2021*). Remarkably, a pharyngeal GRN in each of the two LSOs functions in the rejection rather the acceptance of sucrose (*Joseph et al., 2017*).

In this work, we investigated whether IRs function in pharyngeal GRNs for avoidance of high $Na^+$. We found that IR60b along with co-receptors IR25a and IR76b function in a taste organ in the pharynx for limiting high salt consumption. *Ir60b* is expressed exclusively in a pair of pharyngeal GRNs in the LSO (*Joseph et al., 2017*), and we found that these neurons respond to high $Na^+$. While these *Ir60b* GRNs are narrowly tuned, surprisingly, they have previously been shown to respond to sucrose and glucose (*Joseph et al., 2017*). Introduction of the rat capsaicin receptor, TRPV1 (*Caterina et al., 1997*), into *Ir60b* GRNs induces aversion toward capsaicin, supporting the conclusion that *Ir60b*-positive GRNs are sufficient for instinctive avoidance. To validate these findings further, we used a two-way choice DrosoX assay (*Sang et al., 2021*) to measure actual ingestion levels. We found that the three *Ir* mutants consumed high salt at levels similar to sucrose over an extended period, emphasizing the critical role of this single pair of pharyngeal GRNs in controlling harmful ingestion of high salt.

## Results

### *Ir60b* functions in the repulsion to high salt

To identify potential salt sensors in *Drosophila melanogaster*, we conducted binary food choice assays using 30 *Ir* mutants (*Figure 1A* and *Figure 1—figure supplement 1A*). Through screens in which we gave flies a choice between 2 mM sucrose alone and 2 mM sucrose plus a low, attractive level of salt (50 mM NaCl), we confirmed that *Ir76b* (*Zhang et al., 2013*), *Ir25a*, and *Ir56b* (*Dweck et al., 2022*)

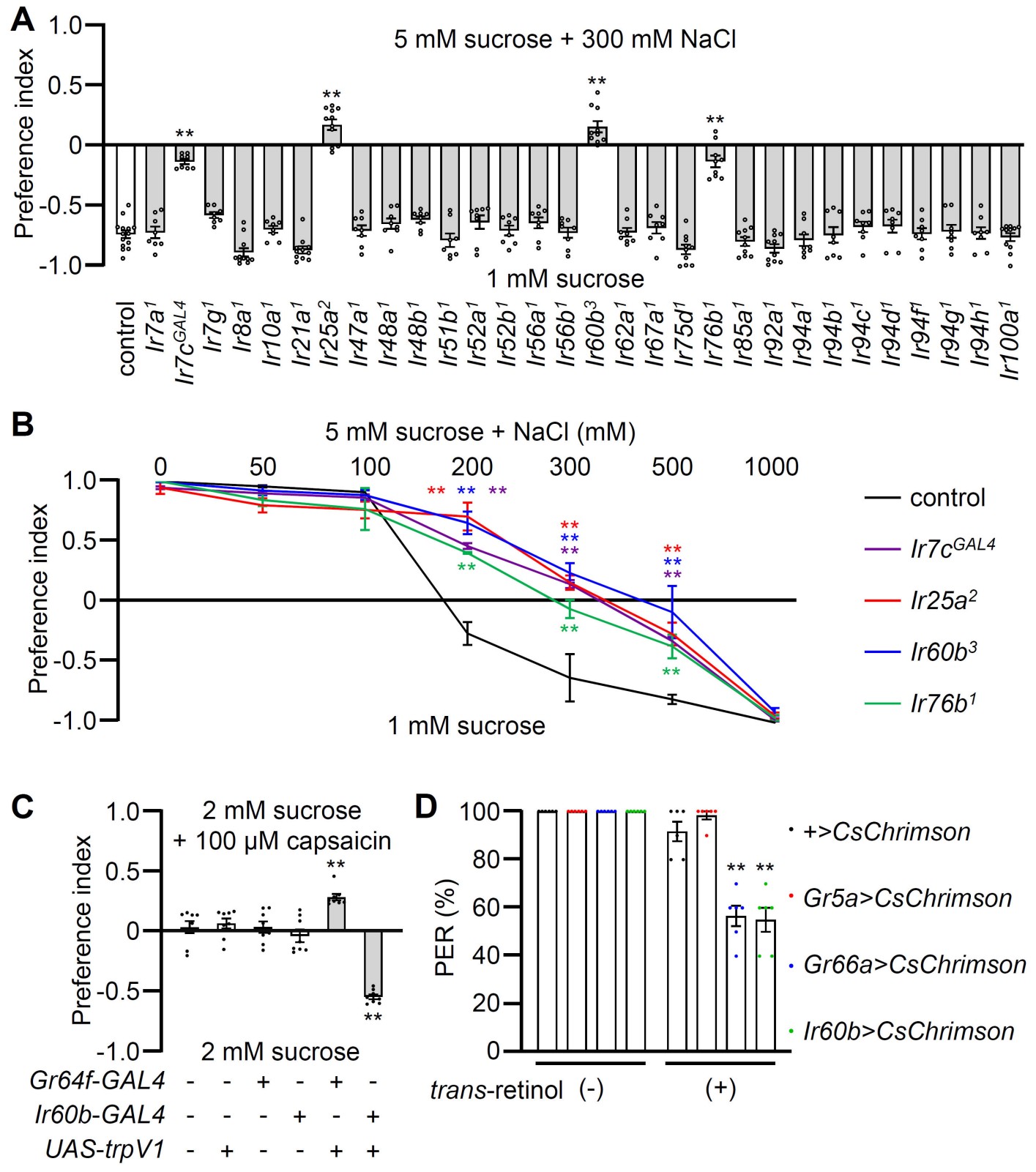

**Figure 1.** Testing requirements for *Ir*s for avoiding high salt-containing food, and chemogenetic and optogenetic control of *Ir60b* gustatory receptor neurons (GRNs). (**A**) Binary food choice assays (1 mM sucrose versus 5 mM sucrose and 300 mM NaCl) comparing 30 *Ir* mutants to the control strain (*w1118*) for high salt avoidance, n=8–12. (**B**) Preferences of the indicated flies for 1 mM sucrose versus 5 mM sucrose and 0–1000 mM NaCl. n=8–12. (**C**) Testing the effects of 100 µM capsaicin after expressing the rat TRPV1 channel (*UAS-trpV1*) either in Class A GRNs or *Ir60b* GRNs under the control

*Figure 1 continued on next page*

*Figure 1 continued*

of the *Gr64f-GAL4* or the *Ir60b-GAL4*, respectively. The flies were given a choice between 2 mM sucrose and 2 mM sucrose plus 100 µM capsaicin. The presence or absence of the various transgenes is indicated by '+' and '-', respectively. n=8. (**D**) Testing the effects of light activation of various classes of GRNs. *UAS-CsChrimson* was expressed in Class A GRNs (driven by the *Gr5a-GAL4*), Class B GRNs (driven by the *Gr66a-GAL4*), or in *Ir60b* GRNs (driven by the *Ir60b-GAL4*). The flies were then simultaneously exposed to red lights (650 nm; WP-5700, 3M, USA) for 5 s while 2% sucrose was applied to labellum and the percent proboscis extension response (PER) was recorded. n=6. Data were compared using single-factor ANOVA coupled with Scheffe's post-hoc test. Statistical significance was compared with the control. Means ± SEMs. **p<0.01.

The online version of this article includes the following figure supplement(s) for figure 1:

**Figure supplement 1.** Requirements for *Irs* for preferring low salt-containing food.

**Figure supplement 2.** Gene structure of the *Ir60b* locus, generation of *Ir60b³* and behavioral defect of *Ir60b¹* in high salt avoidance.

are essential for detecting low salt (***Figure 1—figure supplement 1A***). Moreover, using tip recordings to assay tastant-induced action potentials, we confirmed a previous report (***Dweck et al., 2022***) that loss of *Ir56b* nearly eliminated spikes in S-type and L-type sensilla in response to low salt (***Figure 1—figure supplement 1B***). Using *Ir56b-GAL4* to drive *UAS-mCD8::GFP*, we also confirmed that the reporter was restricted to a subset of Class A GRNs, which were marked with *LexAop-tdTomato* expressed under the control of the *Gr64f-LexA* (***Figure 1—figure supplement 1D–F***). We generated a *UAS-Ir56b* transgene which restored normal frequencies of action potentials in *Ir56b*-expressing GRNs (***Figure 1—figure supplement 1B***). Moreover, ectopic expression of *UAS-Ir56b* in GRNs that typically have minimal responses to low salt caused a large increase in salt-induced action potentials (***Figure 1—figure supplement 1C***).

In our behavioral screen for *Ir* mutants required for avoiding high salt (300 mM NaCl), we found that in addition to *Ir7c*, *Ir25a*, and *Ir76b* as previously described (***McDowell et al., 2022***), *Ir60b* was also required (***Figure 1A***). The *Ir60b* mutant, *Ir60b³*, was generated by removing 768 base pairs, which spanned from 44 base pairs upstream of the predicted transcription start site to the region coding for the N-terminal 241 residues of the 577-amino acid protein (***Figure 1—figure supplement 2A–C***). We verified the impairment in high salt avoidance using a previously described mutant, *Ir60b¹* (***Figure 1—figure supplement 2D***; ***Joseph et al., 2017***). We conducted dose-response behavioral assays using the *Ir60b* mutants, as well as *Ir25a*, *Ir76b*, and *Ir7c* mutants and found that all four exhibited significant deficiencies in avoiding salt concentrations ranging from 200 mM to 500 mM (***Figure 1B***). Nevertheless, all of the mutants exhibited a strong aversion to an extremely high salt concentration (1000 mM), a level twice as concentrated as seawater. This very high level of NaCl could potentially trigger activation of nociceptive neurons, serving as a protective mechanism to prevent potential tissue and organ damage.

## Activation of *Ir60b* neurons inhibits motivation to feed

To investigate whether activation of *Ir60b* GRNs induces aversive behavior, we used both chemogenetic and optogenetic approaches. Capsaicin, a ligand for the mammalian TRPV1 channel (***Caterina et al., 1997***), does not normally elicit responses in flies (***Figure 1C***) as described previously (***Marella et al., 2006***). Therefore, we expressed *UAS-trpV1* under the control of the *Ir60b-GAL4*, and presented the flies with a choice between a 2 mM sucrose and a 2 mM sucrose laced with 100 µM capsaicin. We found that the transgenic flies actively avoided capsaicin (***Figure 1C***), whereas expression of TRPV1 in Class A (sweet) GRNs (*Gr64f-GAL4* and *UAS-trpV1*) induced a preference for capsaicin (***Figure 1C***). These findings support the idea that the activation of *Ir60b* neurons leads to gustatory avoidance.

To further test the proposal that *Ir60b*-positive GRNs elicit aversive behavior, we expressed CsChrimson, a light-activated cation channel (***Klapoetke et al., 2014***) in the *Ir60b* GRN. As controls we drove *UAS-CsChrimson* expression using either the *Gr5a-GAL4* or the *Gr66a-GAL4*. Upon stimulation with red lights and 2% sucrose, nearly all of the control flies (*UAS-CsChrimson* only) or flies expressing *UAS-CsChrimson* in Class A GRNs (*Gr5a-GAL4*) extended their proboscis (***Figure 1D***). In contrast, the proboscis extension response (PER) was notably diminished in flies expressing *UAS-CsChrimson* in the Class B GRNs (*Gr66a-GAL4*) or in the *Ir60b* GRN (*Gr66a-GAL4*; 56.7 ± 4.2% and *Ir60b-GAL4*; 55.0 ± 5.0%, respectively; ***Figure 1D***). These results are fully consistent with a previous study showing optogenetic activation of the *Ir60b* GRN reduces consumption of a sugar (***Joseph et al., 2017***). Together, these findings provide compelling evidence that stimulation of the *Ir60b* GRN induces behavioral aversion.

## *Ir60b* is not required in the labellum to sense high salt

To investigate the physiological responses of labellar sensilla to high salt (300 mM), we conducted tip recordings on each of the 31 sensilla (*Figure 2A*). Five sensilla, including three S-type (S3, S5, and S7) and two L-type (L3 and L4), exhibited the strongest responses to high salt (*Figure 2—figure supplement 1A*). These responses were largely dependent on the IR25a and IR76b co-receptors, as well as IR7c (*Figure 2B and C* and *Figure 2—figure supplement 1B*) as reported (*McDowell et al., 2022*). Interestingly, the *Ir60b³* deletion mutant did not affect the neuronal responses to high salt in labellar taste bristles (*Figure 2B and C*). We inactivated individual GRNs by expressing the inwardly rectifying K⁺ channel (*UAS-Kir2.1*) (*Nitabach et al., 2002*) in Class A GRNs (*Gr64f-GAL4*) (*Dahanukar et al., 2007*), Class B GRNs (*Gr66a-GAL4*) (*Thorne et al., 2004*), Class C GRNs (*ppk28-GAL4*) (*Cameron et al., 2010*), and Class D GRNs (*ppk23-GAL4*) (*Lee et al., 2018*), and confirmed that the aversive behavior and neuronal responses to high salt primarily relied on Class B and D GRNs (*Figure 2D and E*) as described (*Jaeger et al., 2018*).

To examine the gustatory repulsion to high salt that is mediated through the labellum, we conducted PER assays. Starved control and *Ir* mutant flies extend their proboscis when the labellum is lightly touched with a 100 mM sucrose probe (*Figure 2F*). Upon a second sucrose offering, the various fly lines exhibited slightly diminished responses (*Figure 2G*). When we added 300 mM salt to the sucrose, it significantly reduced the PER in the control group (*Figure 2H and I*; first offering 40.9 ± 4.0%; second offering 41.5 ± 3.7%). Both the *Ir25a²* and *Ir76b¹* mutants also exhibited suppressed PERs, but the suppression was not as great as in the control (*Figure 2H and I*). In contrast, high salt reduced the PER by the *Ir60b³* mutant to a similar extent as the control (*Figure 2H and I*; first offering 41.6 ± 6.5%; second offering 47.7 ± 6.7%). This indicates that the labellum of the *Ir60b³* detects 300 mM salt normally, even though the mutant is impaired in avoiding high salt in a two-way choice assay (*Figure 1A*).

## High salt sensor in the pharynx

The observations that *Ir60b* is required for the normal aversion to high salt, but does not appear to function in labellar bristles, raise the possibility that *Ir60b* is required in the pharynx for salt repulsion. *Ir60b* is expressed in the pharynx where it plays a role in limiting sucrose consumption (*Joseph et al., 2017*). *Gr2a* is also expressed in the pharynx and contributes to the repulsion to moderate salt levels (150 mM) (*Kim et al., 2017*). However, the *Gr2aᴳᴬᴸ⁴* mutant displays a normal response to high salt (450 mM) (*Kim et al., 2017*). In our two-way choice assay, which focuses on 300 mM NaCl, we found that salt repulsion displayed by the *Gr2aᴳᴬᴸ⁴* mutant was also indistinguishable from the control (*Figure 2—figure supplement 2*).

To investigate a role for GRNs in the pharynx for high salt (300 mM) repulsion, we conducted tests on the *Poxn* mutant (*Poxn⁷⁰⁻²⁸/Poxnᐃᴹ²²⁻ᴮ⁵*) in which external chemosensory bristles have been converted to mechanosensory sensilla (*Dambly-Chaudière et al., 1992*). The *Poxn* mutants retain GRNs in taste pegs, which are hairless sensilla (*LeDue et al., 2015*). As a result, *Poxn* mutants only possess intact internal gustatory organs, as well as taste pegs. We found that the aversive behavior to high salt was reduced in the *Poxn* mutants relative to the control (*Figure 2J*), consistent with previous studies demonstrating roles for GRNs in labellar bristles in high salt avoidance (*McDowell et al., 2022; Jaeger et al., 2018; Zhang et al., 2013*). However, the diminished high salt avoidance of the *Poxn* mutant was significantly different from the *Poxn;Ir60b³* double mutant, even though the response of *Poxn;Ir60b³* was not significantly different from *Ir60b³* (*Figure 2J*).

## Quantification of increased high salt ingestion in *Ir60b* mutants

In a prior study, it was observed that the repulsion to high salt exhibited by the *Ir60b* mutant was indistinguishable from wild-type (*Joseph et al., 2017*). Specifically, the flies were presented with a drop of liquid (sucrose plus salt) at the end of a probe, and the *Ir60b* mutant flies fed on the food for the same period of time as control flies (*Joseph et al., 2017*). However, this assay did not discern whether or not the volume of the high salt-containing food consumed by the *Ir60b* mutant flies was reduced relative to control flies. Therefore, to assess the volume of food ingested, we used the DrosoX system, which we recently developed (*Figure 3—figure supplement 1A; Sang et al., 2021*). This system consists of a set of five separately housed flies, each of which is exposed to two capillary tubes with different liquid food options. One capillary contained 100 mM sucrose and the other contained

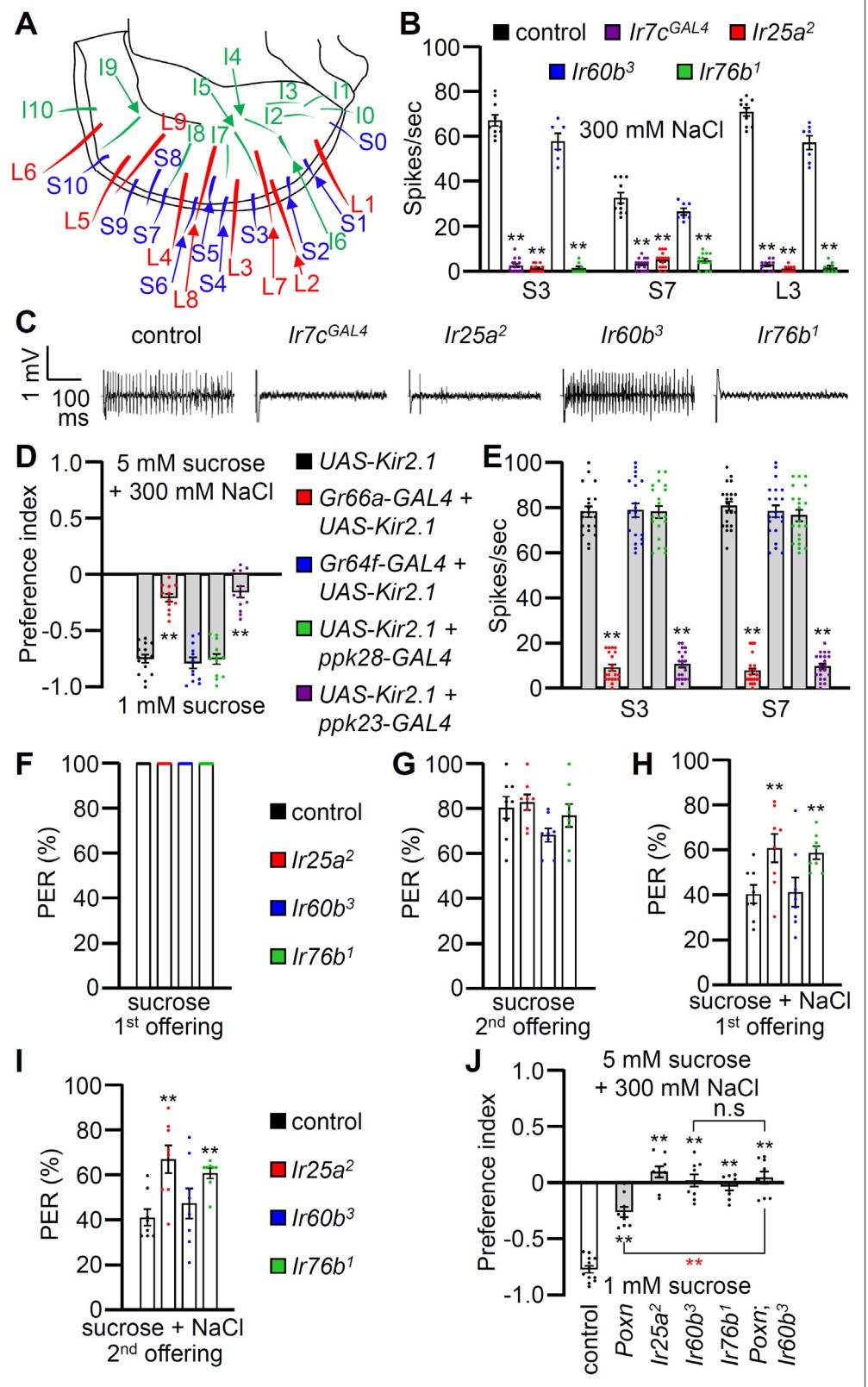

**Figure 2.** Contributions of different classes of gustatory receptor neurons (GRNs) to high salt avoidance. (A) Schematic showing the names of sensilla bristles on the labellum *Weiss et al., 2011*. (B) Tip recordings conducted on S3, S7, and L3 sensilla using 300 mM NaCl and the indicated flies. n=10–16. (C) Representative traces obtained from S3 sensilla. (D) Binary food choice assays (1 mM sucrose versus 5 mM sucrose and 300 mM

*Figure 2 continued on next page*

*Figure 2 continued*

NaCl) after inactivating different classes of GRNs with *UAS-Kir2.1*, driven by the indicated *GAL4* drivers: Class A (*Gr64f*; blue), Class B (*Gr66a*; red), Class C (*ppk28*; green), and Class D (*ppk23*; purple). Significances were determined by comparing to the *UAS-Kir2.1* only control (black). n=12. (**E**) Tip recordings conducted by stimulating S3 and S7 sensilla with 300 mM NaCl from flies with different classes of GRNs inactivated with *UAS-Kir2.1*. See panel (**D**) for legend. n=16–20. (**F–I**) Proboscis extension response (PER) assays performed using the control strain (*w1118*; black) and *Ir25a2* (red), *Ir60b3* (blue), *Ir76b1* (green). n=8–10. (**F**) PER percentages induced by 2% sucrose (first offering). (**G**) PER percentages induced by 2% sucrose (second offering). (**H**) PER percentages induced by 2% sucrose with 300 mM NaCl (first offering). (**I**) PER percentages induced by 2% sucrose with 300 mM NaCl (second offering). (**J**) Binary food choice assays for 300 mM salt avoidance were conducted with the control strain, *Poxn* (*Poxn70-28/PoxnΔM22-B5*), *Ir25a2*, *Ir60b3*, *Ir76b1*, and *Poxn;Ir60b3*. n=9–12. Data were compared using single-factor ANOVA coupled with Scheffe's post-hoc test. Statistical significances compared with the control flies or the *Poxn* mutant are denoted by black and red asterisks, respectively. Means ± SEMs. **p<0.01.

The online version of this article includes the following figure supplement(s) for figure 2:

**Figure supplement 1.** Assaying action potentials induced by different labellar bristles in response to 300 mM salt using tip recordings.

**Figure supplement 2.** Two-way solid food choice assay to assess whether the *Gr2aGAL4* mutant exhibits a deficit in avoidance of high salt.

---

100 mM sucrose mixed with 300 mM NaCl. The volume of food consumed from each capillary was then monitored automatically over the course of 6 hr and recorded on a computer. We found that control flies consuming approximately four times more of the 100 mM sucrose than the sucrose mixed with 300 mM NaCl (*Figure 3A*). In contrast, the *Ir25a*, *Ir60b*, and *Ir76b* mutants consumed approximately twofold less of the sucrose plus salt (*Figure 3A*). Consequently, they ingested similar amounts of the two food options (*Figure 3B*; ingestion index [I.I.]). Thus, while the *Ir60b* mutant and control flies spend similar amounts of time in contact with high salt-containing food when it is the only option (*Joseph et al., 2017*), the mutant consumes considerably less of the high salt food when presented with a sucrose option without salt.

To further investigate the requirement for *Ir25a*, *Ir60b*, and *Ir76b*, we performed genetic rescue experiments. We introduced their respective wild-type cDNAs under the control of their cognate *GAL4* drivers, which resulted in a conversion from salt-insensitive behavior to the salt-sensitive behavior observed in wild-type flies (*Figure 3C–H*). In addition, the defects in the *Ir25a2* and *Ir76b1* mutants were fully rescued by expressing the wild-type *Ir25a* and *Ir76b* transgenes, respectively, in the pharynx using the *Ir60b-GAL4* (*Figure 3I–L*). This suggests that both IR25a and IR76b act as co-receptors in the *Ir60b* GRNs. Furthermore, we investigated whether the expression of *UAS-Ir60b* driven by *Ir25a-GAL4* or *Ir76b-GAL4* could rescue the defects observed in *Ir60b3*. Despite the broad expression of *Ir60b* using these *GAL4* drivers, the *Ir60b* salt ingestion defect was rescued (*Figure 3M and N*).

Next, we addressed whether *Ir60b* is required specifically for regulating ingestion of high salt. To investigate this, we assessed the volumes of caffeine, strychnine, and coumarin consumed by *Ir60b3* flies. We found that the *Ir60b3* mutant displayed similar consumption patterns to the wild-type control flies for these bitter compounds (*Figure 3O and P* and *Figure 3—figure supplement 1B–E*). This is in contrast to the impairments exhibited by the *Gr66aex83* mutant (*Figure 3O and P* and *Figure 3—figure supplement 1B–E*), which displays defects in sensing many bitter chemicals. This indicates that *Ir60b* is involved in regulating the avoidance of high salt ingestion rather than general avoidance responses to toxic compounds. Nevertheless, the role of *Ir60b* in suppressing feeding is not limited to high salt, since *Ir60b* also functions in the pharynx in inhibiting the consumption of sucrose (*Joseph et al., 2017*).

To investigate the aversion induced by high salt in the absence of a highly attractive sugar, such as sucrose, we combined 300 mM salt with 100 mM sorbitol, which is a tasteless but nutritive sugar (*Fujita and Tanimura, 2011*; *Burke and Waddell, 2011*). Using two-way choice assays, we found that the *Ir25a*, *Ir60b*, and *Ir76b* mutants exhibited substantial reductions in high salt avoidance (*Figure 3—figure supplement 2A*). In addition, we performed DrosoX assays using 100 mM sorbitol alone or sorbitol mixed with 300 mM NaCl. Sorbitol alone provoked less feeding than sucrose since it is a tasteless sugar (*Figure 3—figure supplement 2B and C*). Nevertheless, addition of high salt to the sorbitol reduced food consumption (*Figure 3—figure supplement 2B and C*).

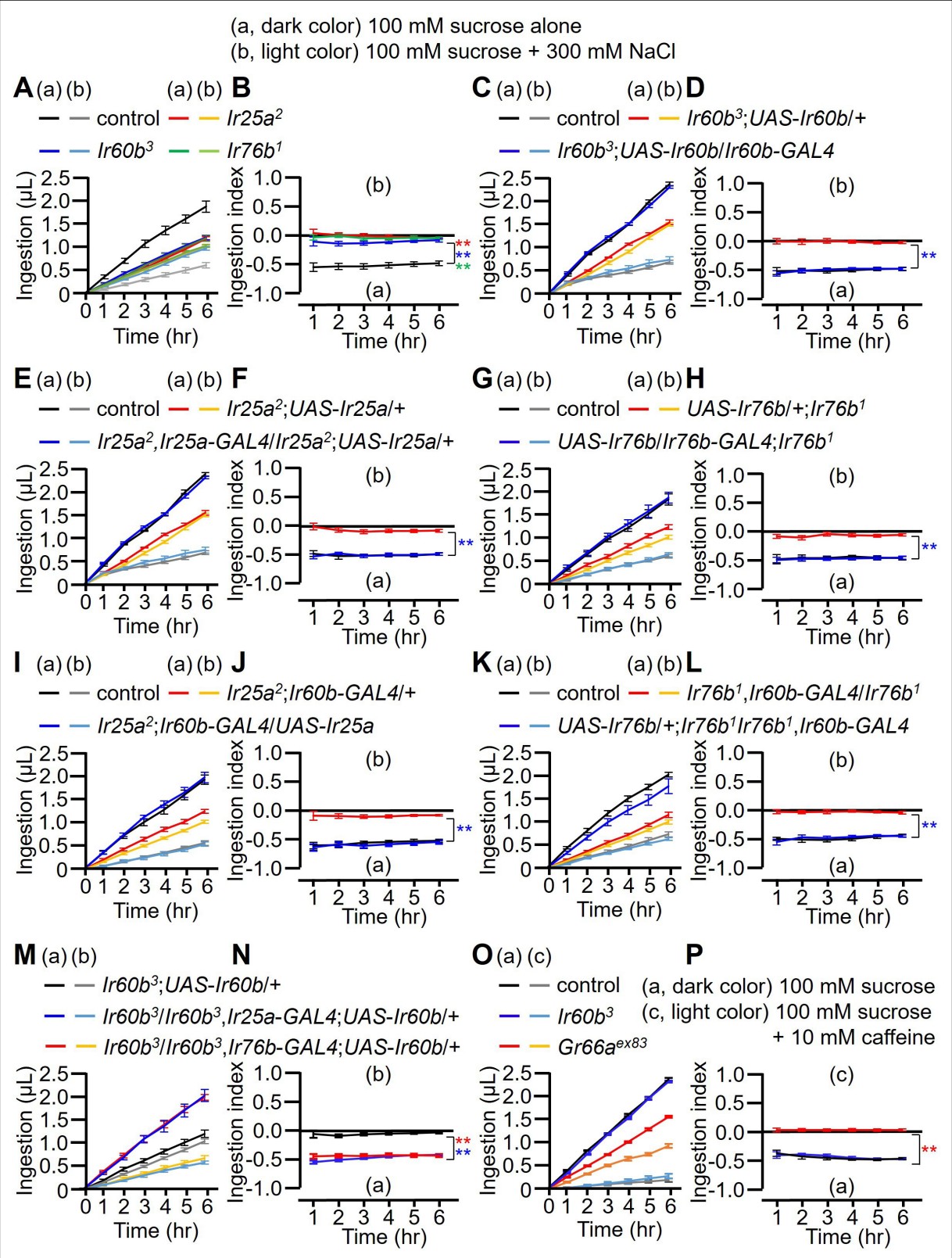

**Figure 3.** Measuring volume of food intake in *Ir* mutants using the DrosoX system. (**A–N**) Each fly was exposed to two capillaries, one of which contained 100 mM sucrose (a), and the other contained 100 mM sucrose and 300 mM NaCl (b). (**O and P**) Each fly was exposed to two capillaries, one of which contained 100 mM sucrose (a), and the other contained 100 mM sucrose and 10 mM caffeine (c). (**A, C, E, G, I, K, M, and O**) Volumes of the two food options consumed by the indicated flies over the course of 6 hr. (**B, D, F, H, I, J, L, N, and P**) Ingestion indexes to indicate the relative

*Figure 3 continued on next page*

*Figure 3 continued*

consumption of the two foods. Ingestion indexes were calculated in each time point using the following equation: [(Ingestion volume of 100 mM sucrose and 300 mM NaCl or 10 mM caffeine) – (Ingestion volume of 100 mM sucrose)]/[(Ingestion volume of 100 mM sucrose and 300 mM NaCl or 10 mM caffeine) + (Ingestion volume of 100 mM sucrose)] n=12. Multiple sets of data were compared using single-factor ANOVA coupled with Scheffe's post-hoc test. Statistical significances were relative to the control and determined for the ingestion indexes only. In all panels, the controls were $w^{1118}$. The colors of the asterisks match the colors of the genotypes in the corresponding panels. Means ± SEMs. **p<0.01.

The online version of this article includes the following figure supplement(s) for figure 3:

**Figure supplement 1.** DrosoX system and measurement of food intake using strychnine and coumarin.

**Figure supplement 2.** Two-way solid food choice assay and DrosoX binary capillary feeding assay using 100 mM sorbitol with or without 300 mM NaCl.

## A single neuron in the LSO depends on *Ir25a*, *IR60b*, and *Ir76b* for responding to both high salt and sucrose

In addition to *Ir60b*, two broadly required *Ir*s (*Ir25a* and *Ir76b*) also function in repulsion to high salt (*Jaeger et al., 2018*; *Zhang et al., 2013*). Moreover, we found that we could rescue the *Ir25a*, *Ir60b*, or *Ir76b* DrosoX phenotypes using the same *Ir60b-GAL4* to drive expression of the cognate wild-type transgenes in the corresponding mutant backgrounds. These findings imply that all three *Ir*s are co-expressed in the *Ir60b* GRN in the pharynx. Therefore, we examined the relative expression patterns of the *Ir60b-GAL4* reporter with the *Ir25a* and *Ir76b* reporters. We observed that the *Ir76b-QF* reporter was expressed in two cells within the LSO, one of which colocalized with the *Ir60b* reporter (*Figure 4A*). Additionally, the expression pattern of the *Ir25a-GAL4* perfectly overlapped with that of *Ir76b-QF* in the LSO (*Figure 4B*). Thus, we suggest that *Ir25a*, *Ir60b*, and *Ir76b* function in the same GRN in the LSO to limit consumption of high salt. We attempted to induce salt activation in the I-type sensilla by ectopically expressing *Ir60b*, under control of the *Gr33a-GAL4*. *Gr33a* is co-expressed with *Gr66a* (*Moon et al., 2009*), which has been shown to be co-expressed with *Ir25a* and *Ir76b* (*Lee et al., 2018*; *Li et al., 2023*). When we performed tip recordings from I5 and I9 sensilla, we did not observe a significant increase in action potentials in response to 300 mM NaCl (*Figure 4—figure supplement 1A*), indicating that ectopic expression of *Ir60b* in combination with *Ir25a* and *Ir76b* is not sufficient to generate a high salt receptor.

To determine whether the *Ir60b* GRN in the LSO is activated by high salt, we examined $Ca^{2+}$ responses in the LSO using *UAS-GCaMP6f*, expressed under the control of each *GAL4* driver. In the wild-type LSO, we identified a single cell that responded to 300 mM NaCl (*Figure 4C*), indicating that the GRN in the LSO that expresses all three reporters responds to high salt. Moreover, this GRN responded robustly to 300–1000 mM $Na^+$ but not to a low level of $Na^+$ (50 mM; *Figure 4E*). We then examined the $Ca^{2+}$ responses in the $Ir25a^2$, $Ir60b^3$, and $Ir76b^1$ mutants, and found that each of them failed to respond to NaCl (*Figure 4D and E*). Additionally, we rescued the deficits in the GCaMP6f responses exhibited by each mutant by expressing a wild-type transgene under control of the corresponding *GAL4* driver (*Figure 4F*). We also tested other $Cl^-$ salts ($CaCl_2$, $MgCl_2$, and KCl) to determine if $Cl^-$ rather than $Na^+$ induced responses in the *Ir60b* GRN. None of these other salts affected these neurons at the 50 mM, 300 mM, and 500 mM concentrations tested (*Figure 4G*). In contrast, NaBr induced GCaMP6f responses (*Figure 4H*). Thus, the *Ir60b* GRN is responsive to $Na^+$ and not $Cl^-$. Due to the effects of NaBr on the *Ir60b* GRN, we used the DrosoX assay to determine whether 300 mM NaBr suppressed ingestion of sucrose. We found that the impact of NaBr on sucrose ingestion was similar to that with NaCl (*Figure 4—figure supplement 1B and C*). We also found that the *Ir60b* GRN did not respond to bitter compounds such as quinine, caffeine, strychnine, lobeline, denatonium, and coumarin at the 5 mM and 50 mM concentrations (*Figure 4I*).

It has been shown previously that *Ir60b* is required in a single GRN in the LSO for suppressing sucrose feeding, and this neuron responds to sucrose (*Joseph et al., 2017*). Therefore, we tested whether the same GRN in the LSO that responds to salt also responds to sucrose. Using GCaMP6f, we found that the *Ir60b* GRN was responsive to sucrose in the LSO of control flies, but not in the *Ir25a*, *Ir60b*, and *Ir76b* mutants (*Figure 4J*). Furthermore, we used GCaMP6f to compare the $Ca^{2+}$ responses exhibited by the *Ir60b* GRN to 100 mM sucrose alone, 300 mM NaCl alone, and a combination of 100 mM sucrose and 300 mM NaCl. We found that the $Ca^{2+}$ responses were significantly higher when we exposed the *Ir60b* GRN to 300 mM NaCl alone, compared with the response to 100 mM sucrose alone (*Figure 4—figure supplement 1D*). However, the GCaMP6f response was not higher when

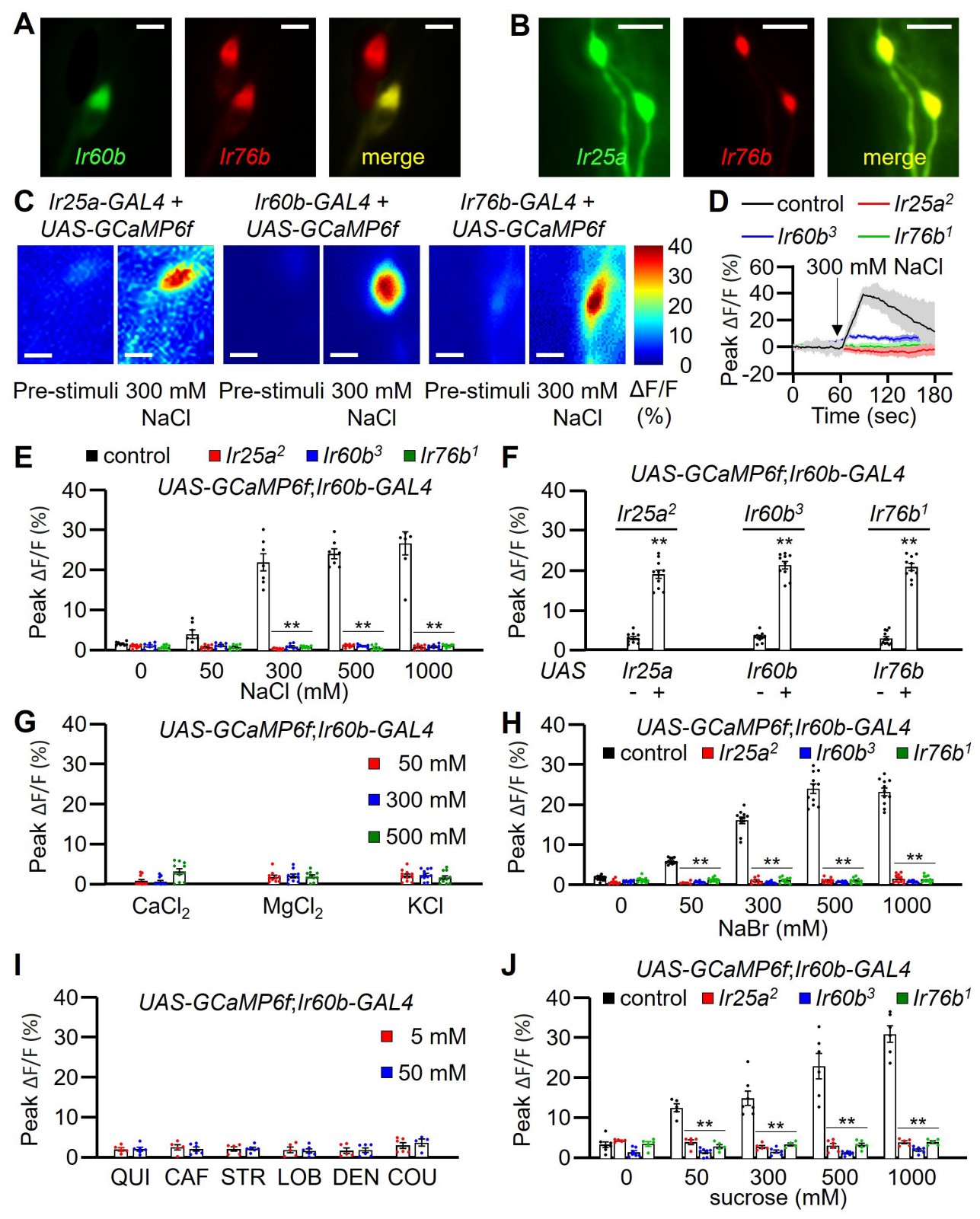

**Figure 4.** GCaMP6f responses of *Ir60b* gustatory receptor neurons (GRNs) to NaCl and other chemicals. (**A**) Relative staining of the *Ir60b* reporter (green, anti-GFP) and the *Ir76b* reporter (red; anti-dsRed) in the labral sense organ (LSO) of *UAS-mCD8::GFP/Ir76b-QF2;Ir60b-GAL4/QUAS-tdTomato* flies. Merge is to the right. (**B**) Relative staining of the *Ir25a* reporter (green, anti-GFP) and the *Ir76b* reporter (red; anti-dsRed) in the LSO of *Ir25a-GAL4/Ir76b-QF2;UAS-mCD8::GFP/QUAS-tdTomato*. Merge is to the right. (**C–J**) Peak GCaMP6f responses (ΔF/F) of *Ir60b* GRNs in flies expressing

*Figure 4 continued on next page*

*Figure 4 continued*

*UAS-GCaMP6f* under control of the indicated *GAL4* driver. (**C**) Heat map images illustrating changes in GCaMP6f fluorescence before and after stimulation with 300 mM NaCl using the indicated flies. (**D**) Sample traces depicting GCaMP6f responses to 300 mM NaCl. The traces are from the indicated flies expressing *UAS-GCaMP6f* driven by the *Ir60b-GAL4*. n=10–14. (**E**) GCaMP6f responses to various concentrations of NaCl in the indicated flies. *UAS-GCaMP6f* was driven by the *Ir60b-GAL4*. n=10–14. (**F**) GCaMP6f responses to 300 mM NaCl in the indicated mutants and in the absence or presence of the corresponding rescue transgene indicated by '-' and '+', respectively. n=8–10. (**G**) GCaMP6f responses to 50 mM, 300 mM, and 500 mM of $CaCl_2$, $MgCl_2$, and KCl in control flies. n=10–14. (**H**) GCaMP6f responses of *Ir60b* GRNs from the indicated flies to various concentrations of NaBr. n=10–14. (**I**) GCaMP6f responses to 5 mM and 50 mM concentrations of bitter compounds (quinine, caffeine, strychnine, lobeline, denatonium, and coumarin). n=8–10. (**J**) GCaMP6f responses to various concentrations of sucrose in *Ir60b* GRNs from the indicated flies. n=10–14. Multiple sets of data were compared using single-factor ANOVA coupled with Scheffe's post-hoc test. Statistical significance compared with the controls. Means ± SEMs. \*\*p<0.01. Scale bars in A–C indicate 5 μm.

The online version of this article includes the following figure supplement(s) for figure 4:

**Figure supplement 1.** Testing whether ectopic expression of *Ir60b* confers responses to 300 mM NaCl, measurement of intake of sucrose plus 300 mM NaBr using the DrosoX assay, and $Ca^{2+}$ response of *Ir60b* gustatory receptor neurons (GRNs).

**Figure supplement 2.** GCaMP6f responses evoked by 300 mM NaCl in *Ir60b* gustatory receptor neurons (GRNs) of control and *Ir7c^GAL4* flies, and relative expression of *Ir60b* and *Ir7c* reporters in the labral sense organ (LSO).

we presented 100 mM sucrose in combination with 300 mM NaCl, compared with the response to 300 mM NaCl alone. We conclude that the same LSO neuron depends on the same three receptors (IR25a, IR60b, and IR76b) for suppressing feeding in response to high salt or to sucrose.

Both Class B and Class D GRNs in labellar bristles respond to high salt (*Montell, 2021*), and the Class D GRNs (marked by *ppk23*) depend on *Ir7c* as well as *Ir25a* and *Ir76b* for responding to high salt (*McDowell et al., 2022*). Consequently, we investigated whether *Ir7c* plays a role in the *Ir60b* GRN in the LSO. We expressed *UAS-GCaMP6f* under control of the *Ir60b-GAL4* in either or *Ir7c^GAL4* mutant background or in a heterozygous (*Ir7c^GAL4/+*) control. We then stimulated the *Ir60b* GRN in the LSO with 300 mM NaCl and found that the responses elicited by the mutant and control were the same (*Figure 4—figure supplement 2A*). Consistent with these findings, we did not detect *Ir7c* reporter expression in the *Ir60b* GRNs (*Figure 4—figure supplement 2B–D*).

## Discussion

The taste organs lining the walls of the pharynx represent the final gatekeepers that flies use to decide whether to continue feeding or to reject the food. We found that activation of a single pharyngeal GRN by high salt depends on three IRs, two of which are widely expressed in other GRNs (*Ir25a* and *Ir76b*) and were previously shown to function in salt taste sensation in external labellar bristles. In addition, we identified a third IR (*Ir60b*) that contributes to salt rejection. Consistent with a previous study (*Joseph et al., 2017*), *Ir60b* is expressed in a single GRN in each of two LSOs lining the pharynx. In addition, we demonstrated that *Ir60b* is co-expressed with *Ir25a* and *Ir76b* in this one GRN in each LSO.

Multiple observations in this study underscore the important role of the *Ir60b* GRN in sensing high $Na^+$ levels and in promoting salt rejection. Mutation of *Ir60b* eliminates the flies ability to reject sucrose laced with high salt over another option with sucrose only. This result is especially notable in view of the observation that flies that are missing all GRNs in labellar bristles in the *Poxn* mutant exhibit a less pronounced defect than the *Ir60b* mutant, since *Poxn* mutant flies still retain some bias for sucrose over sucrose laced with high salt. The observation that one pair of pharyngeal GRNs is sufficient to induce rejection of high salt underscores the profound role of internal taste neurons in protecting flies from ingesting dangerous levels of this mineral. In addition, activation of the *Ir60b* GRN suppresses feeding since optogenetic activation of these neurons effectively suppresses the PER when the flies are presented with a highly attractive tastant (sucrose). Conversely, Yang et al. demonstrated that activation of *Ir60b* neurons can induce the activation of IN1 neurons, potentially leading to heightened feeding (*Yang et al., 2021*). However, our research reveals a specific activation pattern for *Ir60b* neurons. Instead of being generalists, they exhibit specialization for certain sugars, such as sucrose and high salt. As a result, while *Ir60b* GRNs activate IN1 neurons (*Yang et al., 2021*), we posit that there are other neurons in the brain responsible for inhibiting feeding.

A finding that is seemingly conflicting with the optogenetic results is that mutation of *Ir60b* does not reduce the suppression of the PER when the flies are presented with sucrose laced with high Na$^+$ levels. The PER assay under high salt conditions only permits flies to taste the food without allowing ingestion. Consequently, the internal sensor may not exhibit suppression of the PER. In contrast, optogenetic activation has the capability to directly stimulate the internal sensor, leading to the induction of PER suppression. We propose that high Na$^+$ is effective in suppressing the PER by the *Ir60b* mutant since these flies still have functional high salt receptors in aversive GRNs in labellar bristles. In support of this conclusion, loss of either *Ir25a* or *Ir76b* reduces the suppression of the PER by high salt, and this occurs because *Ir76b* and *Ir25a* are required in both high salt activated GRNs in labellar bristles (*Jaeger et al., 2018*; *Zhang et al., 2013*) and in the *Ir60b* GRN in the pharynx.

We also found that the requirement for *Ir60b* appears to be different when performing binary liquid capillary assay (DrosoX), versus solid food binary feeding assays. When we employed the DrosoX assay to test mutants that were missing salt aversive GRNs in labellar bristles but still retained functional *Ir60b* GRNs, the flies behaved the same as wild-type flies (e.g. *Figure 3J and L*). However, using solid food binary assays, *Poxn* mutants, which are missing labellar taste bristles but retain *Ir60b* GRNs (*LeDue et al., 2015*), displayed repulsion to high salt food that was intermediate between control flies and the *Ir60b* mutant (*Figure 2J*). *Poxn* mutants still possess taste pegs (*LeDue et al., 2015*), and these hairless taste organs become exposed to food only when the labial palps open. We suggest that there are high salt-sensitive GRNs associated with taste pegs, which are accessed when the labellum contacts a solid substrate, but not when flies drink from the capillaries used in DrosoX assays. This explanation would also account for the findings that the *Ir60b* mutant is indifferent to 300 mM NaCl in the DrosoX assay (*Figure 3B*), but prefers 1 mM sucrose alone over 300 mM NaCl and 5 mM sucrose in the solid food binary assay (*Figure 1B*). Alternatively, the different behavioral responses might be due to the variation in sucrose concentrations in each of these two assays, which employed 5 mM sucrose in the solid food binary assay, as opposed to 100 mM sucrose in the DrosoX assay. The disparity in attractive valence between these two concentrations of sucrose might consequently impact feeding amount and preference.

Another unresolved question is why does the some pharyngeal GRN respond to sucrose and high Na$^+$. It has been pointed out that since flies prefer over ripe fruit, which has lower sucrose levels that ripe fruit, then activation of the *Ir60b* GRN by sucrose might serve to favor consumption of fruit at an advanced state of ripeness (*Joseph and Heberlein, 2012*). In contrast to sugar, Na$^+$ levels tend to remain constant during the ripening process (*Rop et al., 2010*) and remain at relatively low levels (50–100 mM), which is below the level that is aversive to flies. Thus, in contrast to sucrose, assessing the concentration of Na$^+$ does not aid in the distinction of ripe versus overripe fruit, but may be important in avoiding consuming non-vegetarian sources of food that are high in Na$^+$.

Based on Ca$^{2+}$ imaging results with GCaMP6f, we conclude that the *Ir60b* GRN directly responds to NaCl, and this depends on the presence of *Ir25a*, *Ir60b,* and *Ir76b*. The *Ir60b* GRN responds to high NaCl, but is not stimulated by CaCl$_2$, MgCl$_2$, and KCl. Thus, the *Ir60b* GRN is a Na$^+$ sensor not a Cl$^-$ sensor. In further support of this conclusion, we found that the *Ir60b* GRN also responds to NaBr.

An open question is the subunit composition of the pharyngeal high Na$^+$ receptor, and whether the sucrose/glucose and Na$^+$ receptors in the *Ir60b* GRN are the same or distinct. Our results indicate that the high salt sensor in the *Ir60b* GRN includes IR25a, IR60b, and IR76b since all three IRs are required in the pharynx for sensing high levels of NaCl. I-type sensilla do not elicit a high salt response, and we were unable to induce salt activation in I-type sensilla by ectopically expressing *Ir60b*, under control of the *Gr33a-GAL4*. This indicates that IR25a, IR60b, and IR76b are insufficient for sensing high Na$^+$. The inability to confer a salt response by ectopic expression of *Ir60b* was not due to absence of *Ir25a* and *Ir76b* in *Gr33a* GRNs since *Gr33a* and *Gr66a* are co-expressed (*Moon et al., 2009*), and *Gr66a* GRNs express *Ir25a* and *Ir76b* (*Lee et al., 2018*; *Li et al., 2023*). Thus, the high salt receptor in *Ir60b* GRNs appears to require an additional subunit. Given that Na$^+$ and sugars are structurally unrelated, we suggest that the Na$^+$ and sucrose/glucose receptors do not include the identical set of subunits, or that they activate a common receptor through disparate sites.

Finally, it is remarkable that the single *Ir60b* GRN in the LSO of the pharynx is also stimulated by sucrose and to a lesser extent to glucose (*Joseph et al., 2017*), since this GRN is otherwise very narrowly tuned. We did not detect GCaMP signals upon application any of six bitter tastants tested. The *Ir60b* GRN is also unresponsive to trehalose and glycerol. Based on behavioral experiments,

mutation of *Ir60b* does not impact on consumption of an array of amino acids, low pH, bitter chemicals, and other sugars (*Joseph and Heberlein, 2012*). It is also surprising that the pharyngeal GRN that responds to Na$^+$ is unresponsive to other cations such as Ca$^{2+}$, which is toxic at high levels (*Lee et al., 2018*). The fact that there are relatively few pharyngeal GRNs, yet one is narrowly tuned to sucrose, glucose, and Na$^+$ underscores the critical role of limiting Na$^+$ consumption in flies, which could otherwise lead to dehydration, and dysfunction of many homeostatic processes impacted by excessive levels of Na$^+$ (*Taruno and Gordon, 2023*).

# Materials and methods

## Key resources table

| Reagent type (species) or resource | Designation | Source or reference | Identifiers | Additional information |
|---|---|---|---|---|
| Antibody | Mouse monoclonal anti-GFP | Molecular Probes | Cat # A11120; RPID: AB_221568 | IHC (1:1000) |
| Antibody | Rabbit polyclonal anti-DsRed | Clontech | Cat # 632496; RPID: AB_10013483 | IHC (1:1000) |
| Antibody | Goat polyclonal anti-mouse Alexa Fluro 488 | Invitrogen | Cat # A32723; RRID: AB_2633275 | IHC (1:200) |
| Antibody | Goat polyclonal anti-rabbit Alexa Fluor 568 | Invitrogen | Cat # A11011; RPID: AB_143157 | IHC (1:200) |
| Chemical compound | Sucrose | Sigma-Aldrich | Cat # 9378S | |
| Chemical compound | Tricholine citrate | Sigma-Aldrich | Cat # T0252 | |
| Chemical compound | Sulforhodamine B | Sigma-Aldrich | Cat # 230162 | |
| Chemical compound | Capsaicin | Sigma-Aldrich | Cat # M2028 | |
| Chemical compound | Caffeine | Sigma-Aldrich | Cat # C02750 | |
| Chemical compound | CaCl$_2$ dihydrate | Sigma-Aldrich | Cat # C3881 | |
| Chemical compound | KCl | Sigma-Aldrich | Cat # P9541 | |
| Chemical compound | Quinine | Sigma-Aldrich | Cat # Q1125 | |
| Chemical compound | Strychnine | Sigma-Aldrich | Cat # S8753 | |
| Chemical compound | Lobeline | Sigma-Aldrich | Cat # 141879 | |
| Chemical compound | Denatonium | Sigma-Aldrich | Cat # D5765 | |
| Chemical compound | Coumarin | Sigma-Aldrich | Cat # C4261 | |
| Chemical compound | Brilliant blue FCF | Wako Pure Chemical Industry | Cat # 027-12842 | |
| Chemical compound | Paraformaldehyde | Electron Microscopy Sciences | Cat # 15710 | |
| Chemical compound | NaCl | LPS Solution | Cat # NACL01 | |
| Chemical compound | MgCl$_2$ hexahydrate | SAMCHUN | Cat # M0038 | |
| Chemical compound | NaBr | DUKSAN | Cat # S2531 | |
| Chemical compound | Goat serum, New Zealand origin | Gibco | Cat # 16210064 | |
| Genetic reagent (*Drosophila melanogaster*) | *w$^{1118}$* | Bloomington *Drosophila* Stock Center (BDSC) | BDSC:5905 | |
| Genetic reagent (*Drosophila melanogaster*) | *Ir7a$^1$* | Dr. Y Lee *Rimal et al., 2019* | | |
| Genetic reagent (*Drosophila melanogaster*) | *Ir7g$^1$*: y$^1$w*Mi{y$^{+mDint2}$=MIC}Ir7g$^{MI06687}$ | BDSC | BDSC:42420 | |
| Genetic reagent (*Drosophila melanogaster*) | *Ir7c$^{GAL4}$* | Dr. MD Gordon *McDowell et al., 2022* | | |

*Continued on next page*

*Continued*

| Reagent type (species) or resource | Designation | Source or reference | Identifiers | Additional information |
|---|---|---|---|---|
| Genetic reagent (*Drosophila melanogaster*) | *Ir8a[1]*: w*TI{w[+m*]=TI}Ir8a[1];Bl[1]L[2]/CyO | BDSC | BDSC:23842 | |
| Genetic reagent (*Drosophila melanogaster*) | *Ir10a[1]*: $w^{1118}$Mi{GFP[E.3xP3]=ET1}Ir10a[MB03273] | BDSC | BDSC:41744 | |
| Genetic reagent (*Drosophila melanogaster*) | *Ir21a[1]*: $w^{1118}$;PBac{w[+mC]=PB}Ir21a[c02720] | BDSC | BDSC:10975 | |
| Genetic reagent (*Drosophila melanogaster*) | *Ir25a[2]* | Dr. L Vosshall **Benton et al., 2009** | | |
| Genetic reagent (*Drosophila melanogaster*) | *Ir47a[1]* | Dr. Y Lee **Rimal et al., 2019** | | |
| Genetic reagent (*Drosophila melanogaster*) | *Ir48a[1]*: $w^{1118}$;Mi{GFP[E.3xP3]=ET1}Ir48a[MB09217] | BDSC | BDSC:26453 | |
| Genetic reagent (*Drosophila melanogaster*) | *Ir48b[1]*: $w^{1118}$;Mi{GFP[E.3xP3]=ET1}Ir48b[MB02315] | BDSC | BDSC:23473 | |
| Genetic reagent (*Drosophila melanogaster*) | *Ir51b[1]*: $w^{1118}$;PBac{w[+mC]=PB}row[c00387] Ir51b[c00387] | BDSC | BDSC:10046 | |
| Genetic reagent (*Drosophila melanogaster*) | *Ir52a[1]* | Dr. Y Lee **Rimal et al., 2019** | | |
| Genetic reagent (*Drosophila melanogaster*) | *Ir52b[1]*: $w^{1118}$;Mi{GFP[E.3xP3]=ET1}Ir52b[MB02231]/SM6a | BDSC | BDSC:25212 | |
| Genetic reagent (*Drosophila melanogaster*) | *Ir52c[1]*: $w^{1118}$;Mi{GFP[E.3xP3]=ET1}Ir52c[MB04402] | BDSC | BDSC:24580 | |
| Genetic reagent (*Drosophila melanogaster*) | *Ir56a[1]* | Dr. Y Lee **Rimal et al., 2019** | | |
| Genetic reagent (*Drosophila melanogaster*) | *Ir56b[1]*: $w^{1118}$;Mi{GFP[E.3xP3]=ET1}Ir56b[MB09950] | BDSC | BDSC:27818 | |
| Genetic reagent (*Drosophila melanogaster*) | *Ir56d[1]*: w*;Ir56d[1] | BDSC | BDSC:81249 | |
| Genetic reagent (*Drosophila melanogaster*) | *Ir60b[1]* | Dr. J Carlson **Joseph et al., 2017** | | |
| Genetic reagent (*Drosophila melanogaster*) | *Ir60b[3]* | Dr. Y Lee | In this study | |
| Genetic reagent (*Drosophila melanogaster*) | *Ir62a[1]*: $y^{1}$w*;Mi{y[+mDint2]=MIC}Ir62a[MI00895]Iml1[MI00895]/TM3, Sb[1] Ser[1] | BDSC | BDSC:32713 | |
| Genetic reagent (*Drosophila melanogaster*) | *Ir67a[1]*: $y^{1}$w*;Mi{y[+mDint2]=MIC}Ir67a[MI11288] | BDSC | BDSC:56583 | |
| Genetic reagent (*Drosophila melanogaster*) | *Ir75d[1]*: $w^{1118}$;Mi{GFP[E.3xP3]=ET1}Ir75d[MB04616] | BDSC | BDSC:24205 | |
| Genetic reagent (*Drosophila melanogaster*) | *Ir76b[1]* | Dr. C Montell **Zhang et al., 2013** | | |
| Genetic reagent (*Drosophila melanogaster*) | *Ir85a[1]*: $w^{1118}$;Mi{GFP[E.3xP3]=ET1}Ir85a[MB04613] Pif1A[MB04613] | BDSC | BDSC:24590 | |
| Genetic reagent (*Drosophila melanogaster*) | *Ir92a[1]*: $w^{1118}$;Mi{GFP[E.3xP3]=ET1}Ir92a[MB03705] | BDSC | BDSC:23638 | |
| Genetic reagent (*Drosophila melanogaster*) | *Ir94a[1]* | Dr. Y Lee **Rimal et al., 2019** | | |
| Genetic reagent (*Drosophila melanogaster*) | *Ir94b[1]*:$w^{1118}$;Mi{GFP[E.3xP3]=ET1}Ir94b[MB02190] | BDSC | BDSC:23424 | |
| Genetic reagent (*Drosophila melanogaster*) | *Ir94c[1]* | Dr. Y Lee **Rimal et al., 2019** | | |

*Continued on next page*

*Continued*

| Reagent type (species) or resource | Designation | Source or reference | Identifiers | Additional information |
|---|---|---|---|---|
| Genetic reagent (*Drosophila melanogaster*) | *Ir94d¹*:y¹w\*;Mi{y+mDint2=MIC} Ir94d^MI01659^CG17380^MI01659^ | BDSC | BDSC:33132 | |
| Genetic reagent (*Drosophila melanogaster*) | *Ir94f¹*: y¹w\*;Mi{y+mDint2= MIC}Ir94f^MI00928^ | BDSC | BDSC:33095 | |
| Genetic reagent (*Drosophila melanogaster*) | *Ir94g¹*: w^1118^;Mi{GFP^E.3xP3^= ET1}Ir94g^MB07445^ | BDSC | BDSC:25551 | |
| Genetic reagent (*Drosophila melanogaster*) | *Ir94h¹* | Dr. Y Lee **Rimal et al., 2019** | | |
| Genetic reagent (*Drosophila melanogaster*) | *Ir100a¹*: w^1118^;P{w+mC=EP} Ir100a^G19846^ CG42233^G19846^ | BDSC | BDSC:31853 | |
| Genetic reagent (*Drosophila melanogaster*) | *UAS-mCD8::GFP* | BDSC | BDSC:5137 | |
| Genetic reagent (*Drosophila melanogaster*) | *UAS-mCD8::GFP* | BDSC | BDSC:32184 | |
| Genetic reagent (*Drosophila melanogaster*) | *UAS-Kir2.1* | BDSC | BDSC:6595 | |
| Genetic reagent (*Drosophila melanogaster*) | *UAS-Ir25a* | Dr. Y Lee **Lee et al., 2018** | | |
| Genetic reagent (*Drosophila melanogaster*) | *UAS-Ir60b* | Dr. Y Lee | In this study | |
| Genetic reagent (*Drosophila melanogaster*) | *UAS-Ir76b* | Dr. C Montell **Zhang et al., 2013** | | |
| Genetic reagent (*Drosophila melanogaster*) | *Ir25a-GAL4* | Dr. L Vosshall **Benton et al., 2009** | | |
| Genetic reagent (*Drosophila melanogaster*) | *Ir60b-GAL4* | Dr. C Montell **Joseph et al., 2017** | | |
| Genetic reagent (*Drosophila melanogaster*) | *Ir76b-GAL4* | Dr. C Montell **Zhang et al., 2013** | | |
| Genetic reagent (*Drosophila melanogaster*) | *ppk23-GAL4* | Dr. K Scott **Thistle et al., 2012** | | |
| Genetic reagent (*Drosophila melanogaster*) | *ppk28-GAL4* | Dr. H Amrein **Cameron et al., 2010** | | |
| Genetic reagent (*Drosophila melanogaster*) | *Gr66a-GAL4* | Dr. H Amrein **Thorne et al., 2004** | | |
| Genetic reagent (*Drosophila melanogaster*) | *Gr64f-GAL4* | Dr. A Dahanukar **Lee et al., 2018** | | |
| Genetic reagent (*Drosophila melanogaster*) | *Ir76b-QF* | BDSC | BDSC:51312 | |
| Genetic reagent (*Drosophila melanogaster*) | *QUAS-tdTomato*: y¹w^1118^; P{QUAS-mtdTomato-3xHA}26 | BDSC | BDSC:30005 | |
| Genetic reagent (*Drosophila melanogaster*) | *Poxn^ΔM22-B5^*: y¹w^67c23^; Mi{ET1}Poxn^MB00113^ | BDSC | BDSC:22701 | |
| Genetic reagent (*Drosophila melanogaster*) | *Poxn^70-28^*: Poxn^70^/CyO; twi-Gal4, UAS-2XEGFP | BDSC | BDSC:60688 | |
| Software | Origin Pro Version | Dr. Y Lee | https://www.originlab.com | |
| Software | GraphPad Prism | Dr. Y Lee | https://www.graphpd.com | |
| Software | Autospike 3.1 software | Dr. Y Lee | https://www.syntech.co.za/ | |
| Software | Fiji/ImageJ software | Dr. Y Lee | https://fiji.sc | |
| Software | ZEN lite 2.5 blue | Dr. Y Lee | https://www.zeiss.com/ | |

## Generation of *Ir60b³* and *UAS-Ir60b* lines

The *Ir60b³* mutant was generated by ends-out homologous recombination (**Gong and Golic, 2003**). For generating the construct for injections, approximately two 3 kb genomic fragments were amplified by PCR, and subcloned into *Not*I and *Bam*HI sites of the pw35 vector (**Gong and Golic, 2003**). The resulting mutation deleted the region from –44 to +724 (the A of the ATG initiation codon is defined at +1). The construct was injected into $w^{1118}$ embryos by Best Gene Inc. We outcrossed the mutant to $w^{1118}$ for six generations.

To generate the *UAS-Ir60b* transgenic line, we amplified the full-length *Ir60b* cDNA by reverse transcription polymerase chain reaction using mRNA prepared from whole adult flies and the following primer pair: 5'-GAGAATTCAACTCGAAAATGAGGCGG-3' and 5'-ATGCGGCCGCAATGCTAATTTTG-3'. The *Ir60b* cDNA was subcloned between the *Eco*RI and *Not*I sites of the pUAST vector (**Brand and Perrimon, 1993**), and verified by DNA sequencing. The p*UAS-Ir60b* vector was introduced into $w^{1118}$ embryos by P-element-mediated germline transformation (Korea *Drosophila* Resource Center, Republic of Korea).

## Chemical reagents

The following chemicals were purchased from Sigma-Aldrich (USA): sucrose (CAS No. 57-50-1), tricholine citrate (TCC) (CAS No. 546-63-4), sulforhodamine B (CAS No. 3520-42-1), capsaicin (CAS No. 404-86-4), caffeine (CAS No. 58-08-2), $CaCl_2$ dihydrate (CAS No. 10035-04-8), KCl (CAS No. 7447-40-7), quinine (CAS No. 6119-47-7), strychnine (CAS No. 1421-86-9), lobeline (CAS No. 134-63-4), denatonium (CAS No. 6234-33-6), and coumarin (CAS No. 91-64-5). Brilliant blue FCF (CAS No. 3844-45-9) was purchased from Wako Pure Chemical Industry (Japan). Paraformaldehyde (CAS No. 30525-89-4) was purchased from Electron Microscopy Sciences (USA). NaCl (CAS No. 7647-14-5) was purchased from LPS Solution (Korea). NaBr (CAS No. 7647-15-6) was purchased from DUKSAN (Korea). Goat serum, New Zealand Origin, was purchased from Gibco (USA).

## Binary food choice assay using microtiter dishes

We conducted binary food choice assays as described (**Aryal et al., 2022b**). Briefly, two mixtures were prepared, one of which consisted of 1% agarose, the indicated concentration of NaCl and 5 mM sucrose and red food dye (sulforhodamine B, 0.1 mg/mL). The second mixture contained 1% agarose, 1 mM sucrose, and blue food dye (Brilliant Blue FCF, 0.125 mg/mL). The same phenotypes in **Figure 1B** were verified by swapping the tastants/dye combinations. The two foods were distributed in alternating wells of a 72-well microtiter dish (Cat # 438733, Thermo Fisher Scientific, USA) in a zigzag pattern. 40–50 flies (3—6 days of age) were starved for 18 hr on 1% agarose and transferred to the microtiter dish, which was placed in a dark and humid chamber for 90 min. The flies were then frozen at –20°C, and the colors of their abdomen to determine the number of flies with blue ($N_B$), red ($N_R$), and purple ($N_P$) abdomens. Preference indexes (P.I.s) were calculated using the following equation: $(N_R – N_B)/(N_R + N_B + N_P)$ or $(N_B – N_R)/(N_R + N_B + N_P)$, depending on the specific dye/tastant combinations. A P.I. of –1.0 or 1.0 indicates a complete preference for either 5 mM sucrose with the indicated concentration of NaCl or 1 mM sucrose alone, respectively. A P.I. of 0.0 indicates no preference between the two food alternatives.

## PER assays

PER assays were conducted as described (**Lee et al., 2015**) with minor modifications. Briefly, 20—25 flies (3—6 days of age) were deprived of food for 18—20 hr in vials containing Kimwipe paper wet with tap water. After briefly anesthetizing the flies on ice, they were carefully trapped inside a pipette tip with a volume of 200 µL yellow tip. To expose their heads, the edge of the pipette tip was gently cut using a paper cutter blade. The protruded head and proboscis were used to deliver stimuli. Total 15—20 flies were prepared for the next step. To eliminate any potential biases due to thirst, water was initially provided to the flies with Kimwipe paper until they no longer responded to water. For both the positive control and initial stimulation, a 2% sucrose solution was used. Flies that did not exhibit a response to the sucrose during the initial exposure were excluded from the experiment. The same conditions as the initial exposures were maintained for the second exposure. Therefore, 10—18 flies were selected for the next step. The tastant stimuli, consisting of either 2% sucrose or 2% sucrose mixed with 300 mM NaCl, were presented using Kimwipe paper. We scored the PER as 1.0 with both

complete extensions and partial extensions. PER was calculated using the following equation: (PER flies)/(selected flies). Each test round included 10–18 flies.

## DrosoX binary capillary feeding assay

DrosoX is a recently developed modification of the Expresso technique, which quantifies the amount of feeding by fruit flies (*Yapici et al., 2016*). We conducted DrosoX assays essentially as described (*Sang et al., 2021*). Each sensor bank of the DrosoX system is composed of a printed circuit board housing five Linear Optical Array Sensors (TAOS, TSL1406). Each sensor consists of 768 photodiodes and a microcontroller, establishing a connection to a computer through a Universal Serial Bus port. In the DrosoX setup, when a fly consumes liquid food from a glass capillary, a decrease in the liquid level is identified by a photodiode, enabling the calculation of instantaneous food ingestion. Photodiodes are semiconductor devices, which generate photocurrents upon absorbing light. To ensure light-tight conditions, the sensor bank is enclosed in a box made of black acrylic sheets using precision cutting. A computer reads the electrical signal generated by each photodiode, and the microcontroller (STMicroelectronics, STM32 F103RCBT6) on a development board (Scitech Korea) connected to the DrosoX sensor bank samples the light intensity at each pixel in the array at a rate of 8 Hz. Liquid level readings can be obtained at sample rates ranging from 0.1 to 2 Hz. The data acquisition software records the time vs. liquid level data for multiple sensor banks into a single file using the Hierarchical Data Format (HDF5) (http://www.hdfgroup.org/HDF5/). The obtained data is then analyzed using software (DrosoX_gui) supplied by Scitech Korea.

The conduct the DrosoX assays, we inserted the system in a controlled incubator (25°C, 60% humidity). To quantify ingestion, a mixture comprising 100 mM sucrose and the specified concentration of chemicals was injected into a glass tube (Cat # 53432-706; VWR International, USA) using a syringe (KOVAX-SYRINGE 1 mL 26G; KOREA VACCINE, Korea) and needle (Cat # 90025; Hamilton, Switzerland). DrosoX was equipped with five glass tubes containing a solution, while DrosoXD was equipped with another set of five glass tubes containing different solutions (*Figure 3—figure supplement 1A*). DrosoX and DrosoXD were cross-tested. Each cuvette contained flies (3–6 days of age) and was physically isolated to prevent them from consuming the solution prior to the experiment. Each experiment was conducted for a duration of 6 hr, from 9 AM to 3 PM. The ingestion amount at time X (X hr) was calculated as the difference between the initial solution amount (0 hr) and the solution amount at time X.

The injection index (I.I.) was calculated at each time point using the following equation: (Ingestion volume$_{DrosoX}$ – Ingestion volume$_{DrosoXD}$)/(Ingestion volume$_{DrosoX}$ + Ingestion volume$_{DrosoXD}$) or (Ingestion volume$_{DrosoXD}$ – Ingestion volume$_{DrosoX}$)/(Ingestion volume$_{DrosoXD}$ + Ingestion volume$_{DrosoX}$), depending on the specific tastant combinations. A I.I. of 0.0 indicated no preference based on their ingestion between the two food alternatives.

## Tip recordings to assay tastant-induced action potentials

To measure tastant-induced action potentials, we performed tip recordings as previously described (*Lee et al., 2009*). We immobilized 3- to 6-day-old flies by exposing them to ice. We immobilized a fly by inserting a reference glass electrode filled with Ringer's solution through the back thorax all the way into the proboscis. The recording glass electrode (tip diameter 10–20 µm) contained the NaCl or aversive compounds dissolved in distilled water with 30 mM TCC or 1 mM KCl as the electrolyte. Reference glass electrode and recording glass electrode were created by processing Standard Glass Capillaries (Cat # IB150F-3, World Precision Instruments, USA) with glass puller. The recording electrode was placed over a bristle on the labellum and connected to a pre-amplifier (Taste PROBE, Syntech, Germany), which amplified the signals by a factor of 10 using a signal connection interface box (Syntech) and a 100–3000 Hz band-pass filter. The recorded action potentials were acquired at a sampling rate of 12 kHz and analyzed using Autospike 3.1 software (Syntech). The average frequencies of action potentials (spikes/s) were based on spikes occurring between 50 ms and 550 ms after contact of the recording electrode. The sensilla bristles were defined as described (*Weiss et al., 2011*).

## Immunohistochemistry

We performed immunohistochemistry as previously described (*Lee and Montell, 2013*) with slight modifications. Labella were dissected from 6- to 8-day-old flies and fixed in a solution containing

4% paraformaldehyde (Electron Microscopy Sciences, Cat # 15710) and 0.2% Triton X-100 for 15 min at room temperature. The tissues were then washed three times with PBST (1× PBS and 0.2% Triton X-100), bisected using a razor blade, and incubated in blocking buffer (0.5% goat serum in 1× PBST) for 30 min at room temperature. To detect the target protein, primary anti-bodies (mouse anti-GFP; Molecular Probes, Cat # A11120; diluted 1:1000 and rabbit anti-dsRed; Clontech, Cat # 632496; diluted 1:1000) were added to fresh blocking buffer and incubated with the samples overnight at 4°C. The tissues were then washed three times with PBST, incubated with the secondary antibodies (goat anti-mouse Alexa Fluor 488; Invitrogen, Cat # A32723; diluted 1:200 and goat anti-rabbit Alexa Fluor 568; Invitrogen, Cat # A11011; diluted 1:200) for 4 hr at 4°C, washed three times with PBST, and placed in 1.25× PDA mounting buffer (containing 37.5% glycerol, 187.5 mM NaCl, and 62.5 mM Tris pH 8.8). The signals were visualized using a Leica Stellaris 5 confocal microscope.

## Ex vivo $Ca^{2+}$ imaging using GCaMP6f

Ex vivo $Ca^{2+}$ imaging was performed as previously described (*Inagaki et al., 2014*) with slight modifications using 6- to 8-day-old flies expressing *UAS-GCaMP6f* driven by *Ir25a-GAL4*, *Ir60b-GAL4*, or *Ir76b-GAL4*, which were incubated at 25°C, under 12 hr light/12 hr dark cycles and 50–60%, humidity. 0.5% low melting agarose was applied to a confocal dish (Cat # 102350, SPL LIFE SCIENCE, Korea). After solidification of the low melting agarose, a blade was used to cut and create a shallow well for sample fixation. Fly heads were carefully decapitated using sharp razor blades, followed by excising a small portion of the extended proboscis to facilitate tastant access to the pharyngeal organs. The tissue sample was then carefully fixed in an inverted position in the pre-prepared well.

Adult hemolymph (AHL: 108 mM NaCl, 5 mM KCl, 8.2 mM $MgCl_2$, 2 mM $CaCl_2$, 4 mM $NaHCO_3$, 1 mM $NaH_2PO_4$, and 5 mM HEPES pH 7.5) was used as *Drosophila* imaging saline. After recording 1 min as a pre-stimulus (20 μL AHL), we imaged the $Ca^{2+}$ dynamics following the application of a specific tastant (fivefold higher concentration in 5 μL AHL). GCaMP6f fluorescence was observed using a fluorescence microscope (Axio Observer 3; Carl Zeiss) with a 20× objective, specifically focusing on the relevant area of the pharynx. Videos were recorded at a speed of 2 frames/s. Neuronal fluorescent activity changes were recorded for 5 min following stimulus application. We did not use a perfusion system to wash the stimulus. Fiji/ImageJ software (https://fiji.sc) was used to measure fluorescence intensities. A region of interest (ROI) was drawn around the cell bodies, and the Time-Series Analyzer Plugin, developed by Balaji, J. (https://imagej.nih.gov/ij/plugins/time-series.html), was used to measure the average intensity for the ROIs during each frame. The average pre-stimulation value before chemical stimulation was calculated. ΔF/F (%) was determined using the formula $(F_{max}-F_0)/F_0 \times 100\%$, where $F_0$ represents the baseline value of GCaMP6f averaged for 10 frames immediately before stimulus application, and $F_{max}$ is the maximum fluorescence value observed after stimulus delivery.

## Statistical analyses

Error bars indicate the standard error of the means (SEMs), while the dots represent the number of trials conducted for the experiment. To compare multiple datasets, we used single-factor ANOVA coupled with Scheffe's analysis as a post-hoc test. Pairwise comparisons were conducted using unpaired Student's *t*-tests. Statistical significance is denoted by asterisks (*$p<0.05$, **$p<0.01$). We performed all statistical analyses using Origin Pro 8 for Windows (ver. 8.0932; Origin Lab Corporation, USA).

## Acknowledgements

This work was supported by grants to YL from the National Research Foundation of Korea (NRF) funded by the Korea government (MIST) (NRF-2021R1A2C1007628) and Biomaterials Specialized Graduate Program through the Korea Environmental Industry and Technology Institute (KEITI) funded by the Ministry of Environment (MOE), and grants to CM from the National Institute on Deafness and other Communication Disorders (NIDCD), R01-DC007864 and R01-DC016278. SD, BS, and DN were supported by the Global Scholarship Program for Foreign Graduate Students at Kookmin University in Korea.

## Additional information

### Funding

| Funder | Grant reference number | Author |
|---|---|---|
| National Research Foundation of Korea | NRF-2021R1A2C1007628 | Youngseok Lee |
| National Institute on Deafness and Other Communication Disorders | R01-DC007864 | Craig Montell |
| National Institute on Deafness and Other Communication Disorders | R01-DC016278 | Craig Montell |
| Korea Environmental Industry and Technology Institute | | Youngseok Lee |

The funders had no role in study design, data collection and interpretation, or the decision to submit the work for publication.

### Author contributions

Jiun Sang, Data curation, Formal analysis, Validation, Investigation, Visualization, Methodology, Writing – original draft, Writing – review and editing; Subash Dhakal, Data curation, Formal analysis, Validation, Investigation, Visualization, Methodology; Bhanu Shrestha, Data curation, Investigation, Visualization; Dharmendra Kumar Nath, Conceptualization, Data curation, Formal analysis, Investigation; Yunjung Kim, Resources, Formal analysis, Investigation, Methodology; Anindya Ganguly, Methodology; Craig Montell, Supervision, Funding acquisition, Writing – original draft, Project administration, Writing – review and editing; Youngseok Lee, Conceptualization, Supervision, Funding acquisition, Investigation, Writing – original draft, Project administration, Writing – review and editing

### Author ORCIDs

Jiun Sang (iD) http://orcid.org/0000-0002-0824-8428
Craig Montell (iD) http://orcid.org/0000-0001-5637-1482
Youngseok Lee (iD) http://orcid.org/0000-0003-0459-1138

Reviewer #1 (Public review): https://doi.org/10.7554/eLife.93464.3.sa1
Reviewer #2 (Public review): https://doi.org/10.7554/eLife.93464.3.sa2
Reviewer #3 (Public review): https://doi.org/10.7554/eLife.93464.3.sa3
Author response https://doi.org/10.7554/eLife.93464.3.sa4

## Additional files

### Supplementary files

• MDAR checklist

### Data availability

Source data for all figures contained in the manuscript and SI have been deposited in figshare.

The following dataset was generated:

| Author(s) | Year | Dataset title | Dataset URL | Database and Identifier |
|---|---|---|---|---|
| Sang J | 2023 | Source data - Sang et al., 2023. | https://doi.org/10.6084/m9.figshare.23939394 | figshare, 10.6084/m9.figshare.23939394 |

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
