## [Editor Report · eLife assessment]

This **valuable** study on the molecular and cellular mechanisms of ingestion avoidance of high salt in insects is focused in scope, but the authors present **convincing** evidence that a specific subset of gustatory receptors in a pair of pharyngeal taste neurons are necessary and sufficient for avoiding ingestion of high salt during feeding. This work will be of interest to *Drosophila* neuroscientists interested in taste coding and feeding behavior.

---

## [Referee Report · Reviewer #1 (Public review)]

Summary:

In this manuscript, Sang et al. proposed a pair of IR60b-expressing pharyngeal neurons in *Drosophila* use IR25a, IR76b, and IR60b channels to detect high Na+ and limit its consumption. Some of the key findings that support this thesis are: (1) animals that lacked any one of these channels - or with their IR60b-expressing neurons selectively silenced - showed much reduced rejection of high Na+, but restored rejection when these channels were reintroduced back in the IR60b neurons; (2) animals with TRPV artificially expressed in their IR60b neurons rejected capsaicin-laced food whereas WT did not; (3) IR60b-expressing neurons exhibited increased Ca2+ influx in response to high Na+ and such response went away when animals lacked any of the three channels.

The experiments were thorough and well designed and further improved after revision. The results are compelling and support the main claim. The development and the use of the DrosoX two-choice assay put forward for a more quantitative and automatic/unbiased assessment for ingestion volume and preference.

---

## [Referee Report · Reviewer #2 (Public review)]

Summary:

In this paper, Sang et al. set out to identify gustatory receptors involved in salt taste sensation in *Drosophila melanogaster*. In a two-choice assay screen of 30 Ir mutants, they identify that Ir60b is required for avoidance of high salt. In addition, they demonstrate that activation of Ir60b neurons is sufficient for gustatory avoidance using either optogenetics or TRPV1 to specifically activate Ir60b neurons. Then, using tip recordings of labellar gustatory sensory neurons and proboscis extension response behavioral assays in Ir60b mutants, the authors demonstrate that Ir60b is dispensable for labellar taste neuron responses to high salt and the suppression of proboscis extension by high salt. Since external gustatory receptor neurons (GRNs) are not implicated, they look at Poxn mutants, which lack external chemosensory sensilla but have intact pharyngeal GRNs. High salt avoidance was reduced in Poxn mutants but was still greater than Ir60b mutants, suggesting that pharyngeal gustatory sensory neurons alone are sufficient for high salt avoidance. The authors use a new behavioral assay to demonstrate that Ir60b mutants ingest a higher volume of sucrose mixed with high salt than control flies do, suggesting that the action of Ir60b is to limit high salt ingestion. Finally, they identify that Ir60b functions within a single pair of gustatory sensory neurons in the pharynx, and that these neurons respond to high salt but not bitter tastants.

Strengths:

A great strength of this paper is that it rigorously corroborates previously published studies that have implicated specific Irs in salt taste sensation. It further introduces a new role for Ir60b in limiting high salt ingestion, demonstrating that Ir60b is necessary and sufficient for high salt avoidance and convincingly tracing the action of Ir60b to a particular subset of gustatory receptor neurons. Overall the authors have achieved their aim by identifying a new gustatory receptor involved in limiting high salt ingestion. They use rigorous genetic, imaging, and behavioral studies to achieve this aim, often confirming a given conclusion with multiple experimental approaches. They have further done a great service to the field by replicating published studies and corroborating the roles of a number of other Irs in salt taste sensation.

---

## [Referee Report · Reviewer #3 (Public review)]

Sang et al. successfully demonstrate that a set of single sensory neurons in the pharynx of *Drosophila* promotes avoidance of food with high salt concentrations, complementing previous findings on Ir7c neurons with an additional internal sensing mechanism. The experiments are well-conducted and presented, convincingly supporting their important findings and extending the understanding of internal sensing mechanisms.

The authors convincingly demonstrate the avoidance phenotype using different behavioral assays, thus comprehensively analyzing different aspects of the behavior. The experiments are straightforward and well-contextualized within existing literature.

---

## [Author Response]

The following is the authors’ response to the original reviews.

**Public Reviews:**

**Reviewer #1 (Public review):**
Summary:In this manuscript, Sang et al. proposed a pair of IR60b-expressing pharyngeal neurons in *Drosophila* use IR25a, IR76b, and IR60b channels to detect high Na+ and limit its consumption. Some of the key findings that support this thesis are: (1) animals that lacked any one of these channels - or with their IR60b-expressing neurons selectively silenced - showed much reduced rejection of high Na+, but restored rejection when these channels were reintroduced back in the IR60b neurons; (2) animals with TRPV artificially expressed in their IR60b neurons rejected capsaicin-laced food whereas WT did not; (3) IR60b-expressing neurons exhibited increased Ca2+ influx in response to high Na+ and such response went away when animals lacked any of the three channels.Strengths:The experiments were thorough and well designed. The results are compelling and support the main claim. The development and the use of the DrosoX two-choice assay put forward for a more quantitative and automatic/unbiased assessment for ingestion volume and preference.Weaknesses:There are a few inconsistencies with respect the the exact role by which IR60b neurons limit high salt consumption and the contribution of external (labellar) high-salt sensors in regulating high salt consumption. These weaknesses do not significantly impact the main conclusion, however.
**Reviewer #2 (Public review):**
Summary:In this paper, Sang et al. set out to identify gustatory receptors involved in salt taste sensation in *Drosophila melanogaster*. In a two-choice assay screen of 30 Ir mutants, they identified that Ir60b is required for avoidance of high salt. In addition, they demonstrate that activation of Ir60b neurons is sufficient for gustatory avoidance using either optogenetics or TRPV1 to specifically activate Ir60b neurons. Then, using tip recordings of labellar gustatory sensory neurons and proboscis extension response behavioral assays in Ir60b mutants, the authors demonstrate that Ir60b is dispensable for labellar taste neuron responses to high salt and the suppression of proboscis extension by high salt. Since external gustatory receptor neurons (GRNs) are not implicated, they look at Poxn mutants, which lack external chemosensory sensilla but have intact pharyngeal GRNs. High salt avoidance was reduced in Poxn mutants but was still greater than Ir60b mutants, suggesting that pharyngeal gustatory sensory neurons alone are sufficient for high salt avoidance. The authors use a new behavioral assay to demonstrate that Ir60b mutants ingest a higher volume of sucrose mixed with high salt than control flies do, suggesting that the action of Ir60b is to limit high salt ingestion. Finally, they identify that Ir60b functions within a single pair of gustatory sensory neurons in the pharynx, and that these neurons respond to high salt but not bitter tastants.Strengths:A great strength of this paper is that it rigorously corroborates previously published studies that have implicated specific Irs in salt taste sensation. It further introduces a new role for Ir60b in limiting high salt ingestion, demonstrating that Ir60b is necessary and sufficient for high salt avoidance and convincingly tracing the action of Ir60b to a particular subset of gustatory receptor neurons. Overall, the authors have achieved their aim by identifying a new gustatory receptor involved in limiting high salt ingestion. They use rigorous genetic, imaging, and behavioral studies to achieve this aim, often confirming a given conclusion with multiple experimental approaches. They have further done a great service to the field by replicating published studies and corroborating the roles of a number of other Irs in salt taste sensation. An aspect of this study that merits further investigation is how the same gustatory receptor neurons and Ir in the pharynx can be responsible for regulating the ingestion of both appetitive (sugar) and aversive tastants (high salt).

A previous report published in eLife from John Carlson’s lab (Joseph et al, 2017) showed that the Ir60b GRN in the pharynx responds to sucrose resulting in sucrose repulsion. Thus, stimulation of this pharyngeal GRN results in gustatory avoidance only, not both attraction and avoidance. (lines 205-207)

Weaknesses:There are several weaknesses that, if addressed, could greatly improve this work.(1) The authors combine the results and discussion but provide a very limited interpretation of their results. More discussion of the results would help to highlight what this paper contributes, how the authors interpret their results, and areas for future study.

We agree and have now separated the Results and Discussion, and in so doing have greatly expanded discussion of the results.

(2) The authors rename previously studied populations of labellar GRNs to arbitrary letters, which makes it difficult to understand the experiments and results in some places. These GRN populations would be better referred to according to the gustatory receptors they are known to express.

One of the corresponding authors (Craig Montell) introduced this alternative GRN nomenclature in a review in 2021: Montell, C. (*Drosophila* sensory receptors—a set of molecular Swiss Army Knives. Genetics 217, 1-34) (Montell, 2021). We are not fans of referring to different classes of GRNs based on the receptors that they express since it is not obvious which receptors to use. For example, the GRNs that respond to bitter compounds all express multiple GR co-receptors. The same is true for the GRNs that respond to sugars. The former system of referring to GRNs simply as sugar, bitter, salt and water GRNs is also not ideal since the repertoire of chemicals that stimulates each class is complex. For example, the Class A GRNs (formerly sugar GRNs) are also activated by low Na+, glycerol, fatty acids, and acetic acid, while the B GRNs (former bitter GRNs) are also stimulated by high Na+, acids, polyamines, and tryptophan. In addition, there are five classes of GRNs. At first mention of the Class A—E GRNs, we mention the most commonly used former nomenclature of sugar, bitter, salt and water GRNs. In addition, for added clarify, we now also include a mention of one of the receptors that mark each class. (lines 51-59)

(3) The conclusion that GRNs responsible for high salt aversion may be inhibited by those that function in low salt attraction is not well substantiated. This conclusion seems to come from the fact that overexpression of Ir60b in salt attraction and salt aversion sensory neurons still leads to salt aversion, but there need not be any interaction between these two types of sensory neurons if they act oppositely on downstream circuits.

We did not make this claim.

(4) The authors rely heavily on a new Droso-X behavioral apparatus that is not sufficiently described here or in the previous paper the authors cite. This greatly limits the reader's ability to interpret the results.

We expanded the description of the apparatus in the Droso-X assay section of the Materials and Methods. (lines 588-631)

**Reviewer #3 (Public review):**
Summary:Sang et al. successfully demonstrate that a set of single sensory neurons in the pharynx of *Drosophila* promotes avoidance of food with high salt concentrations, complementing previous findings on Ir7c neurons with an additional internal sensing mechanism. The experiments are well-conducted and presented, convincingly supporting their important findings and extending the understanding of internal sensing mechanisms. However, a few suggestions could enhance the clarity of the work.Strengths:The authors convincingly demonstrate the avoidance phenotype using different behavioral assays, thus comprehensively analyzing different aspects of the behavior. The experiments are straightforward and well-contextualized within existing literature.Weaknesses:DiscussionWhile the authors effectively relate their findings to existing literature, expanding the discussion on the surprising role of Ir60b neurons in both sucrose and salt rejection would add depth. Additionally, considering Yang et al. 2021's (https://doi.org/10.1016/j.celrep.2021.109983) result that Ir60b neurons activate feeding-promoting IN1 neurons, the authors should discuss how this aligns with their own findings.

Yang et al. demonstrated that the activation of Ir60b neurons can trigger the activation of IN1 neurons akin to pharyngeal multimodal (PM) neurons, potentially leading to enhanced feeding (Yang et al, 2021). However, our research reveals a specific pattern of activation for Ir60b neurons. Instead of being generalists, they are specialized for certain sugars, such as sucrose and high salt. Consequently, while Ir60b GRNs activate IN1 neurons, we contend that there are other neurons in the brain responsible for inhibiting feeding. (lines 412-417)

Lines 187: The discussion primarily focuses on taste sensillae outside the labellum, neglecting peg-type sensillae on the inner surface. Clarification on whether these pegs contribute to the described behaviors and if the Poxn mutants described also affect the pegs would strengthen the discussion.

We added the following to the Discussion section. “We also found that the requirement for Ir60b appears to be different when performing binary liquid capillary assay (DrosoX), versus solid food binary feeding assays. When we employed the DrosoX assay to test mutants that were missing salt aversive GRNs in labellar bristles but still retained functional Ir60b GRNs, the flies behaved the same as wild-type flies (e.g. Figure 3J and 3L). However, using solid food binary assays, Poxn mutants, which are missing labellar taste bristles but retain Ir60b GRNs (LeDue et al, 2015), displayed repulsion to high salt food that was intermediate between control flies and the Ir60b mutant (Figure 2J). Poxn mutants retain taste pegs (LeDue et al., 2015), and these hairless taste organs become exposed to food only when the labial palps open. We suggest that there are high-salt sensitive GRNs associated with taste pegs, which are accessed when the labellum contacts a solid substrate, but not when flies drink from the capillaries used in DrosoX assays. This explanation would also account for the findings that the Ir60b mutant is indifferent to 300 mM NaCl in the DrosoX assay (Figure 3B), but prefers 1 mM sucrose alone over 300 mM NaCl and 5 mM sucrose in the solid food binary assay (Figure 1B).”. (lines 430-444)

In line 261 the authors state: "We attempted to induce salt activation in the I-type sensilla by ectopically expressing Ir60b, similar to what was observed with Ir56b 8; however, this did not generate a salt receptor (Figures S6A)"An obvious explanation would be that these neurons are missing the identified necessary co-receptors Ir76b and Ir25a. The authors should discuss here if the Gr33a neurons they target also express these co-receptors, if yes this would strengthen their conclusion that an additional receptor might be missing.

We clarified this point in the Discussion section as follows, “An open question is the subunit composition of the pharyngeal high Na+ receptor, and whether the sucrose/glucose and Na+ receptors in the Ir60b GRN are the same or distinct. Our results indicate that the high salt sensor in the Ir60b GRN includes IR25a, IR60b and IR76b since all three IRs are required in the pharynx for sensing high levels of NaCl. I-type sensilla do not elicit a high salt response, and we were unable to induce salt activation in I-type sensilla by ectopically expressing Ir60b, under control of the Gr33a-GAL4. This indicates that IR25a, IR60b and IR76b are insufficient for sensing high Na+. The inability to confer a salt response by ectopic expression of Ir60b was not due to absence of Ir25a and Ir76b in Gr33a GRNs since Gr33a and Gr66a are co-expressed (Moon et al, 2009), and Gr66a GRNs express Ir25a and Ir76b (Li et al, 2023). Thus, the high salt receptor in Ir60b GRNs appears to require an additional subunit. Given that Na+ and sugars are structurally unrelated, we suggest that the Na+ and sucrose/glucose receptors do not include the identical set of subunits, or that that they activate a common receptor through disparate sites”. (lines 464-477)

MethodsThe description of the Droso-X assay seems to be missing some details. Currently, it is not obvious how the two-choice is established. Only one capillary is mentioned, I assume there were two used? Also, the meaning of the variables used in the equation (DrosoX and DrosoXD) are not explained.

We expanded the description of the apparatus in the Droso-X assay section of the Materials and Methods. (lines 588-631)

The description of the ex-vivo calcium imaging prep. is unclear in several points:(1) It is lacking information on how the stimulus was applied (was it manually washed in? If so how was it removed?).

We expanded the description of the apparatus in the ex vivo calcium imaging section of the Materials and Methods. (lines 682-716)

(2) The authors write: "A mild swallow deep well was prepared for sample fixation." I assume they might have wanted to describe a "shallow well"?

We deleted the word “deep.”.(line 691)

(3) "...followed by excising a small portion of the labellum in the extended proboscis region to facilitate tastant access to pharyngeal organs." It is not clear to me how one would excise a small portion of the labellum, the labellum depicts the most distal part of the proboscis that carries the sensillae and pegs. Did the authors mean to say that they cut a part of the proboscis?

Yes. We changed the sentence to “…followed by excising a small portion of the extended proboscis to facilitate tastant access to the pharyngeal organs.”.(lines 693)-695

**Recommendations for the authors:**

**Reviewer #1 (Recommendations For The Authors):**
In this manuscript, Sang et al. proposed a pair of IR60b-expressing pharyngeal neurons in *Drosophila* use IR25a, IR76b, and IR60b channels to detect high Na+ and limit its consumption. Some of the key findings that support this thesis are: (1) animals that lacked any one of these channels - or with their IR60b-expressing neurons selectively silenced - showed much reduced rejection of high Na+, but restored rejection when these channels were reintroduced back in the IR60b neurons; (2) animals with TRPV artificially expressed in their IR60b neurons rejected capsaicin-laced food whereas WT did not; (3) IR60b-expressing neurons exhibited increased Ca2+ influx in response to high Na+ and such response went away when animals lacked any of the three channels. In general, I find the collective evidence presented by the authors convincing. But I feel the MS can benefit from having a discussion session and a few simple experiments. Below I listed some inconsistencies I hope the authors can address or at least discuss.

We have now added a Discussion section, and expanded the discussion.

(1) The role of IR60b neurons on suppressing PER appeared inconsistent. On the one hand, optogenetic activation of these neurons suppressed PER (Fig 1D), on the other hand, IR60b mutants were as competent to suppress PER in response to high salt as WT (Fig 2G). Are pharyngeal neurons expected to modulate PER? It might be worth including a retinal-free or genotype control to ascertain the PER suppression exhibited by IR60b>CsChrimson is genuine.

Please note that Figure 2G is now Figure 2H.

Our interpretation is that activation of aversive GRNs by high salt either in labellar bristles or in the pharynx is sufficient to inhibit repulsion to high salt. Consistent with this conclusion, optogenetic activation of Ir60b GRNs, which are specific to the pharynx, is sufficient to reduce the PER to sucrose containing food (Figure 1D). However, mutation of Ir60b has no impact on the PER to sucrose plus high (300 mM) NaCl since the high-salt activated GRNs in labellar bristles are not impaired by the Ir60b mutation. In contrast, Ir25a and Ir76b are required in both labellar bristles and in the pharynx to reject high salt. As a consequence, mutation of either Ir25a or Ir76b impairs the repulsion to high salt. Thus, there is no inconsistency between the optogenetics and PER results. We clarified this point in the Discussion section. In terms of controls for IR60b>CsChrimson, we show that UAS-CsChrimson alone or UAS-CsChrimson in combination with the Gr5a driver has no impact on the PER (Figure 1D). In addition, we now include a retinal free control (Figure 1D). These findings provide the key genetic controls and are described in the Results section. (lines 167-170)

(2) The role of labellar high-salt sensors in regulating salt intake appeared inconsistent. On the one hand, they appeared to have a role in limiting high salt consumption because poxn mutants were significantly more receptive to high salt than WT (Fig. 2J). On the other hand, selectively restoring IR76b or IR25a in only the IR60b neurons in these mutants - thus leaving the labellar salt sensors still defective - reverted the flies to behave like WT when given a choice between sucrose vs. sucrose+high salt (Fig 3J, L).

We now offer an explanation for these seemingly conflicting results in the Discussion section. When we employed the DrosoX assay with mutants with functional Ir60b GRNs, but were missing salt aversive GRNs in labellar bristles, the flies behaved the same as control flies (e.g. Figure 3J and L). However, using solid food binary assays, Poxn mutants, which are missing labellar taste bristles but retain Ir60b GRNs (LeDue et al., 2015), display aversion high salt food intermediate between control and Ir60b mutant flies (Figure 2J). Poxn mutants retain taste pegs (LeDue et al., 2015), which are exposed to food substrates only when the labial palps open. We suggest that the taste pegs harbor high salt sensitive GRNs, and they may be exposed to solid substrates, but not to the liquid in capillary tubes used in the DrosoX assays. This explanation would also account for the findings that the Ir60b mutant is indifferent to 300 mM NaCl in the DrosoX assay (Figure 3B), but prefers 1 mM sucrose alone over 300 mM NaCl and 5 mM sucrose in the solid food binary assay (Figure 1B). (lines 433-444)

(3) The behavior sensitivity of IR60b mutant to high salt again appeared somewhat inconsistent when assessed in the two different choice assays. IR60b mutant flies were indifferent to 300 mM NaCl when assayed with DrosoX (Fig 3A, B) but were clearly still sensitive to 300 mM NaCl when assayed with "regular" assay - they showed much reduced preference for 5 mM sucrose over 1 mM sucrose when the 5 mM sucrose was adulterated with 300 mM NaCl (Fig 1B).

The explanation provided above may also account for the findings that the Ir60b mutant is indifferent to 300 mM NaCl in the DrosoX assay (Figure 3B), but not when selecting between 300 mM NaCl and 5 mM sucrose versus 1 mM sucrose in the solid food binary assay (Figure 1B). Alternatively, the different behavioral responses might be due to the variation in sucrose concentrations in each of these two assays, which employed 5 mM sucrose in the solid food binary assay, as opposed to 100 mM sucrose in the DrosoX assay. This disparity in attractive valence between these two concentrations of sucrose might consequently impact feeding amount and preference. This point is now also included in the Discussion section. (lines 441-449)

(4) Given the IR60b neurons exhibited clear IR60b/IR25a/IR76b-dependent sucrose sensitivity, too, I am curious how the various mutant animals behave when given a choice between 100 mM sorbitol vs. 100 mM sorbitol + 300 mM NaCl, a food choice assay not complicated by the presence of sucrose. Similarly, I am curious if the Ca2+ response of IR60 neurons differs significantly when presented with 100 mM sucrose vs. when presented with 100 mM sucrose + 300 mM NaCl. In principle, the magnitude for the latter should be significantly larger than the former as animals appeared to be capable of discriminating these two choices solely relying on their IR60b neurons.

To investigate the aversion induced by high salt in the absence of a highly attractive sugar, such as sucrose, we combined 300 mM salt with 100 mM sorbitol, which is a tasteless but nutritive sugar (Burke & Waddell, 2011; Fujita & Tanimura, 2011). Using two-way choice assays, we found that the Ir25a, Ir60b, and Ir76b mutants exhibited substantial reductions in high salt avoidance (Figure 3—figure supplement 2A). In addition, we performed DrosoX assays using 100 mM sorbitol alone, or sorbitol mixed with 300 mM NaCl. Sorbitol alone provoked less feeding than sucrose since it is a tasteless sugar (Figure 3—figure supplement 2B and C). Nevertheless, addition of high salt to the sorbitol reduced food consumption (Figure 3—figure supplement 2B and C). (lines 300-308)

We also conducted a comparative analysis of the Ca2+ responses within the Ir60b GRN, examining its reaction to various stimuli, including 100 mM sucrose alone, 300 mM NaCl alone, and a combination of 100 mM sucrose and 300 mM NaCl. We found that the Ca2+ responses were significantly higher when we exposed the Ir60b GRN to 300 mM NaCl alone, compared with the response to 100 mM sucrose alone (Figure 4—figure supplement 1D). However, the GCaMP6f responses was not higher when we presented 100 mM sucrose with 300 mM NaCl, compared with the response to 300 mM NaCl alone (Figure 4—figure supplement 1D). (lines 360-367)

Minor issues(1) The labels of sucrose concentration on Figure 2D were flipped.

This has been corrected.

(2) The phrasing of the sentence that begins in line 196 (i.e., "This suggests the internal sensor ...") is not as optimal.

We changed the sentence to, “We found that the aversive behavior to high salt was reduced in the Poxn mutants relative to the control (Figure 2J), consistent with previous studies demonstrating roles for GRNs in labellar bristles in high salt avoidance (Jaeger et al, 2018; McDowell et al, 2022; Zhang et al, 2013).”. (lines 217-219)

(3) In Line 231, I am not sure why the authors think ectopic expressing IR60b in labellar neurons would allow them to become activated by Na+. It seems highly unlikely to me, especially given IR60b also plays a role in sensing sugar.

We added the following paragraph to the Discussion addressing this point, “An open question is the subunit composition of the pharyngeal high Na+ receptor, and whether the sucrose/glucose and Na+ receptors in the Ir60b GRN are the same or distinct. Our results indicate that the high salt sensor in the Ir60b GRN includes IR25a, IR60b and IR76b since all three IRs are required in the pharynx for sensing high levels of NaCl. I-type sensilla do not elicit a high salt response, and we were unable to induce salt activation in I-type sensilla by ectopically expressing Ir60b, under control of the Gr33a-GAL4. This indicates that IR25a, IR60b and IR76b are insufficient for sensing high Na+. The inability to confer a salt response by ectopic expression of Ir60b was not due to absence of Ir25a and Ir76b in Gr33a GRNs since Gr33a and Gr66a are co-expressed (Moon et al., 2009), and Gr66a GRNs express Ir25a and Ir76b (Li et al., 2023). Thus, the high salt receptor in Ir60b GRNs appears to require an additional subunit. Given that Na+ and sugars are structurally unrelated, we suggest that the Na+ and sucrose/glucose receptors do not include the identical set of subunits, or that that they activate a common receptor through disparate sites.”. (lines 464-477)

**Reviewer #2 (Recommendations for the authors):**
Line 41, acutely excessive salt ingestion can lead to death, not just health issues

We now state that, “consumption of excessive salt can contribute to various health issues in mammals, including hypertension, osteoporosis, gastrointestinal cancer, autoimmune diseases, and can lead to death.”. (lines 41-43)

Line 46, delete the comma after flies

Done. (line 47)

Lines 51-56: This description is unnecessarily confusing and does not cite proper sources. Renaming these GRNs arbitrarily can only create confusion, plus this description lacks nuance. If E GRNs are Ir94e positive, this description is out of date. Furthermore, If D GRNs are ppk23 and Gr66a positive then they will respond to both bitter and high salt.Papers to consult: https://elifesciences.org/articles/37167, https://doi.org/10.1016/j.cell.2023.04.038

We have now added citations. We prefer the A—E nomenclature, which was introduced in a 2021 Genetics review by one of the authors of this manuscript (Montell) (Montell, 2021) since naming different classes of GRNs on the basis of markers or as sweet, bitter, salt and water GRNs is misleading and an oversimplification. We cite the Genetics 2021 review, and for added clarity include both types of former names (markers and sweet, bitter, salt and water). Class D GRNs are not marked by Gr66a. The eLife reference cited above provided the initial rationale for stating that Class E GRNs are marked by Ir94e and activated by low salt. According to the Taisz et al reference (Cell 2023), the Class E GRNs, which are marked by Ir94e, are also activated by pheromones, which we now mention (Taisz et al, 2023). (lines 51-59)

Line 62, E GRNs are not required for low salt behaviors

We do not state that E GRNs are required for low salt behaviors, only that they sense low Na+ levels. (line 58)

Line 70-81 - Great deal of emphasis on labellar GRNs but then no mention of how pharyngeal GRNs fit into categories A-E

We devote the following paragraph to pharyngeal GRNs. We do not mention how they fit in with the A—E categories because it is not clear.

“In addition to the labellum and taste bristles on other external structures, such as the tarsi, fruit flies are endowed with hairless sensilla on the surface of the labellum (taste pegs), and three internal taste organs lining the pharynx, the labral sense organ (LSO), the ventral cibarial sense organ (VCSO), and the dorsal cibarial sense organ (DCSO), which also function in the decision to keep feeding or reject a food (Chen & Dahanukar, 2017, 2020; LeDue et al., 2015; Nayak & Singh, 1983; Stocker, 1994). A pair of GRNs in the LSO express a member of the gustatory receptor family, Gr2a, and knockdown of Gr2a in these GRNs impairs the avoidance to slightly aversive levels of Na+ (Kim et al, 2017). Pharyngeal GRNs also promote the aversion to bitter tastants, Cu2+, L-canavanine, and bacterial lipopolysaccharides (Choi et al, 2016; Joseph et al., 2017; Soldano et al, 2016; Xiao et al, 2022). Other pharyngeal GRNs are stimulated by sugars and contribute to sugar consumption (Chen & Dahanukar, 2017; Chen et al, 2021; LeDue et al., 2015). Remarkably, a pharyngeal GRN in each of the two LSOs functions in the rejection rather the acceptance of sucrose (Joseph et al., 2017).”. (lines 74-89)

Line 89, aversive  aversion

We changed this part.

Line 90, gain of aversion capsaicin avoidance suggests they are sufficient for avoidance, not essential for avoidance.

We changed “essential” to “sufficient.”. (line 100)

Line 104, what are you recording from here? Labellar or pharyngeal GRNs

We added “S-type and L-type sensilla” to the sentence. (line 119)

Line 107, How are A GRNS marked with tdTomato? It is important to mention how you are defining A GRNs.

We modified the sentence as follows: “Using Ir56b-GAL4 to drive UAS-mCD8::GFP, we also confirmed that the reporter was restricted to a subset of Class A GRNs, which were marked with LexAop-tdTomato expressed under the control of the Gr64f-LexA (Figure 1—figure supplement 1D—F).”. (lines 120-123)

Line 124, should read "concentrated as sea water."

We made the change. (line 142)

Line 125, I am not sure what is meant by "alarm neurons"

We changed “additional pain or alarm neurons” to “nociceptive neurons.”. (line 144)

Line 141, Are you definitely A GRNs as only labellar GRNs, i.e. the Gr5a-GAL4 pattern with labellar plus few pharyngeal GRNs? Or are the defining it as Gr64f-GAL4 (i.e. labellar plus many pharyngeal GRNs)

We refer to the Class A—E GRNs as labellar GRNs. Therefore, in this instance, we removed the reference to A GRNs and B GRNs, and simply mention the drivers that we used (Gr5a-GAL4 and Gr66a-GAL4) to express UAS-CsChrimson. The modified sentence is, “As controls we drove UAS-CsChrimson under control of either the Gr5a-GAL4 or the Gr66a-GAL4.”. (lines 51-59, 160-161)

Line 180, labellar hairs labellar taste bristles

We made the change. (line 204)

Line 190, possess only  only possess

We made the change. (line 216)

Line 202, Should this read increased?

Yes. We changed “reduced” to “increased.”. (line 225)

Line 206, The information provided here and in reference 47 was not sufficient for me to understand how the Droso-X system works and whether it has been validated. Better diagrams and much more description is required for the reader to understand this system and assess its validity

We now explain that the DrosoX “system consists of a set of five separately housed flies, each of which is exposed to two capillary tubes with different liquid food options. One capillary contained 100 mM sucrose and the other contained 100 mM sucrose mixed with 300 mM NaCl. The volume of food consumed from each capillary is then monitored automatically over the course of 6 hours and recorded on a computer.”. (lines 238-243)

Line 218-219, It would be helpful to expand on this to explain how the previous paper detected no difference. Is this because the contact time with the food is the same but the rate of ingestion is slower?

Yes. This is correct. We now clarify this point by stating that, “In a prior study, it was observed that the repulsion to high salt exhibited by the Ir60b mutant was indistinguishable from wild-type (Joseph et al., 2017). Specifically, the flies were presented with drop of liquid (sucrose plus salt) at the end of a probe, and the Ir60b mutant flies fed on the food for the same period of time as control flies (Joseph et al., 2017). However, this assay did not discern whether or not the volume of the high salt-containing food consumed by the Ir60b mutant flies was reduced relative to control flies. Therefore, to assess the volume of food ingested, we used the DrosoX system, which we recently developed (Figure 3—figure supplement 1A) (Sang et al, 2021). This system consists of a set of five separately housed flies, each of which is exposed to two capillary tubes with different liquid food options. One capillary contained 100 mM sucrose and the other contained 100 mM sucrose mixed with 300 mM NaCl. The volume of food consumed from each capillary was then monitored automatically over the course of 6 hours and recorded on a computer. We found that control flies consuming approximately four times more of the 100 mM sucrose than the sucrose mixed with 300 mM NaCl (Figure 3A). In contrast, the Ir25a, Ir60b, and Ir76b mutants consumed approximately two-fold less of the sucrose plus salt (Figure 3A). Consequently, they ingested similar amounts of the two food options (Figure 3B; ingestion index). Thus, while the Ir60b mutant and control flies spend similar amounts of time in contact with high salt-containing food when it is the only option (Joseph et al., 2017), the mutant consumes considerably less of the high salt food when presented with a sucrose option without salt.”. (lines 226-251)

Lines 231-235, Is this evidence for this, that Ir60b expression in the Ir25a or Ir76b pattern will induce high salt responses in the labellum? You should elaborate on this to clearly state what you mean rather than implying it. I do not think that overexpression of one Ir is enough evidence for this sweeping conclusion.

We agree. We eliminated this point. (lines 227-232)

Lines 261-263, Please elaborate here, how did you target the I-type sensilla and where are these neurons? So they already express Ir76b and Ir25a?

We now explain in the Results that, “We attempted to induce salt activation in the I-type sensilla by ectopically expressing Ir60b, under control of the Gr33a-GAL4. Gr33a is co-expressed with Gr66a (Moon et al., 2009), which has been shown to be co-expressed Ir25a and Ir76b (Li et al., 2023). When we performed tip recordings from I7 and I10 sensilla, we did not observe a significant increase in action potentials in response to 300 mM NaCl (Figure 4—figure supplement 1A), indicating that ectopic expression of Ir60b in combination with Ir25a and Ir76b is not sufficient to generate a high salt receptor.”. (lines 324-330)

Lines 300-303, The discussion needs to be greatly expanded. What is the proposed mechanism by which the same neurons/receptors can inhibit sucrose and high salt feeding? What is the author's interpretation of what this study adds to our understanding of taste aversion?

We have now added a Discussion section and greatly expanded the discussion.

**Reviewer #3 (Recommendations for the authors):**
In line 73 there is a typo in "esophagus"

We changed this part.

In line 331, the use of a mixture of sucrose and "saponin" seems to be a mistake; "NaCl" is likely intended.

We made the correction. (lines 546 and 640)

On several occasions, the authors refer to the pharynx as a taste organ (for example 1st sentence of the abstract). I am not sure this is correct, the actual pharyngeal taste organs are the LSO, DSCO, and VSCO which are located in the pharynx.

We made the corrections. (lines 24, 90, 92, 93, and 356)

In line 155 the authors refer to Ir25a and Ir76b as "broadly tuned". I think it is not correct to refer to co-receptors this way, I'd suggest to just call them co-receptors.

We made the correction. (lines 177-178)

In line 182, stating "Gr2a is also expressed in the proboscis" is unclear. Clarify whether it refers to sensillae, pharyngeal taste organs, etc.

We clarified it refers to pharyngeal taste organs. (lines 206-207)

Line 253: "These finding imply that all three Irs are coexpressed in the pharynx." "The pharynx" is very unspecific, did the authors mean to say "the same neuron"?

We now clarify by saying “in the Ir60b GRN in the pharynx.”. (line 317)

Figures & LegendsI found it confusing that the same color scale is being reused for different panels with different meanings repeatedly and in inconsistent ways. For example in Figure 2, red and blue are being used for Ir25a² mutants, while blue is also being used for Gr64f-Gal4 and S type sensilla. It is also not easily visible nor mentioned in the caption which of the 3 color scales presented belong to which panels.

We modified the colors in the figures so that they are used in a consistent way. We now also define the colors in the legends.

In Figure 2 F-I, indicating the stimulus sequence in each panel would enhance clarity.The color scale in Figure 3 could benefit from explicit explanations of different shades in the caption for easier interpretation.For example: "The ingestion of (a, dark color) 100 mM sucrose alone and (b, light color) in combination with 300 mM"

We made the suggested modification.

In Figure 4a the authors highlight that Ir76b and Ir25a label 2 neurons in the LSO. Did the imaging in 4c also capture the second cell, and if so did it respond to their stimulation?

No, the focal plane differs, and the signal in Figure 4C is considerably weaker compared to the immunohistochemistry shown in Figure 4A. Notably, the other neuron did not exhibit a response to NaCl.

In Figure 4f a legend for the color scale is missing, or the color might not be necessary at all. Also, the asterisks seem to be shifted to the right.

We fixed the shifted asterisks and eliminated the color.

Figure 4i is mislabeled 4f

We made the correction.